# Spatiotemporal variations of the $\delta(O_2/N_2)$, $CO_2$ and $\delta(APO)$ in the troposphere over the Western North Pacific

Shigeyuki Ishidoya[1], Kazuhiro Tsuboi[2], Yosuke Niwa[3], Hidekazu Matsueda[2], Shohei Murayama[1], Kentaro Ishijima[2], and Kazuyuki Saito[4]

[1]National Institute of Advanced Industrial Science and Technology (AIST), Tsukuba 305-8569, Japan,
[2]Meteorological Research Institute, Tsukuba, Japan, Tsukuba305-0052, Japan,
[3]National Institute for Environmental Studies, Tsukuba 305-8506, Japan
[4]Japan Meteorological Agency, Tokyo, Japan, Tokyo 105-8431, Japan

*Correspondence to*: Shigeyuki Ishidoya (s-ishidoya@aist.go.jp)

**Abstract.** We analyzed air samples collected onboard a cargo aircraft C-130 over the western North Pacific from May 2012 to March 2020 for atmospheric $\delta(O_2/N_2)$ and $CO_2$ amount fraction. Observations were corrected for significant artificial fractionation of $O_2$ and $N_2$ caused by thermal diffusion during the air sample collection by using the simultaneously-measured $\delta(Ar/N_2)$. The observed seasonal cycles of the $\delta(O_2/N_2)$ and atmospheric potential oxygen ($\delta(APO)$) varied nearly in opposite phase to that of the $CO_2$ amount fraction at all latitudes and altitudes. Seasonal amplitudes of $\delta(APO)$ decreased with latitude from 34 to 25° N, as well as with increasing altitude from the surface to 6 km by 50–70 %, while those of $CO_2$ amount fraction decreased by less than 20%. By comparing the observed values with the simulated $\delta(APO)$ and $CO_2$ amount fraction values generated by an atmospheric transport model, we found that the seasonal $\delta(APO)$ cycle in the middle troposphere was modified significantly by a combination of the northern and southern hemispheric seasonal cycles due to the inter-hemispheric mixing of air. The simulated $\delta(APO)$ underestimated the observed interannual variation in $\delta(APO)$ significantly, probably due to the interannual variation in the annual mean air-sea $O_2$ flux. Interannual variation in $\delta(APO)$ driven by the net marine biological activities, obtained by subtracting the assumed solubility-driven component of $\delta(APO)$ from the total variation, indicated a clear influence on annual net sea-to-air marine biological $O_2$ flux during El Niño and net air-to-sea flux during La Niña. By analyzing the observed secular trends of $\delta(O_2/N_2)$ and $CO_2$ amount fraction, global average terrestrial biospheric and oceanic $CO_2$ uptakes for the period 2012–2019 were estimated to be (1.8 ± 0.9) and (2.8 ± 0.6) Pg a$^{-1}$ (C equivalents), respectively.

## 1 Introduction

Atmospheric $O_2/N_2$ ratios have been observed since the early 1990s, for the primary application of constraining the marine and terrestrial exchange of $CO_2$ (Keeling and Shertz, 1992). For this purpose, observations of the $O_2/N_2$ ratio have been carried out at many surface stations and on commercial cargo ships (e.g., Bender et al., 2005; Manning and Keeling, 2006; Tohjima et al., 2008, 2019; Goto et al., 2017). The $O_2/N_2$ ratio varies in opposite phase with $CO_2$ amount fraction due to the

terrestrial biosphere exchanges and fossil fuel combustion, of which respective $O_2:CO_2$ exchange ratios (oxidative ratio (OR), $-\Delta y(O_2)\Delta y(CO_2)^{-1}$) are about 1.1 and 1.4, respectively (Keeling, 1988; Severinghaus, 1995), where $y$ stands for the dry amount fraction of gas, as recommended by the IUPAC Green Book (2007). By using the OR value of 1.1 for terrestrial biospheric activities, Atmospheric Potential Oxygen (APO) is defined by $y(APO) = y(O_2) +1.1y(CO_2)$ (Stephens et al., 1998). While APO is conserved for terrestrial biospheric activities, the air-sea exchange of $O_2$ is much faster than that of $CO_2$ since the air-sea $CO_2$ exchange is highly suppressed by the carbonate buffer system in seawater (e.g. Keeling et al., 1993). Therefore, APO can be used to evaluate air-sea $O_2$ fluxes associated with marine biological and physical processes (e.g. Nevison et al., 2012).

Aircraft observation serves as a useful platform for measuring altitude-dependent APO driven by spatially-integrated air-sea $O_2$ fluxes at the surface. Aircraft observations of the $O_2/N_2$ ratio have been conducted in the past (e.g. Sturm et al., 2005; Steinbach, 2010; Ishidoya et al., 2008a, 2012, 2014; van der Laan et al., 2014; Bent, 2014; Morgan et al., 2019; Birner et al., 2020; Stephens et al., 2018, 2021). Sturm et al. (2005) observed a vertical gradient and seasonal cycle in the $O_2/N_2$ ratio in the altitude range of 0.8-3.1 km over Perthshire, United Kingdom, for the period 2003-2004. Longer term observations of the tropospheric $O_2/N_2$ ratio have also been carried out by Ishidoya et al. (2012) and van der Laan et al. (2014). They conducted aircraft observations at altitudes of 2, 4 and above 8 km over Japan during 1999-2010 and at altitudes of 0.1 and 3 km over western Russia during 1998-2008, respectively, and provided additional evidence of seasonal cycles and secular changes in the tropospheric $O_2/N_2$ ratio. However, there were uncertainties associated with artificial fractionations of $O_2$ and $N_2$ in Ishidoya et al. (2012) and van der Laan et al. (2014). $Ar/N_2$ ratio and/or stable isotopic ratios can be used to evaluate natural and artificial molecular-diffusive fractionations of $O_2$ and $N_2$ (e.g. Kawamura et al., 2006; Ishidoya et al., 2013), however Ishidoya et al. (2012) and van der Laan et al. (2014) did not observe them.

Ishidoya et al. (2014) and Stephens et al. (2021) observed $O_2/N_2$ and $Ar/N_2$ simultaneously, and were able to correct for thermally-diffusive artificial fractionation of $O_2/N_2$, by using coefficients of 3.54 and 3.77 of $Ar/O_2$, respectively. By using the corrected $O_2/N_2$ ratio, Ishidoya et al. (2014) were able to observe spatiotemporal variations in the $O_2/N_2$ ratio from the surface to the middle troposphere over the western North Pacific around Japan, on monthly scheduled flights, for the period May 2012 to April 2013. Similarly, Stephens et al. (2021) were able to provide a better picture of much wider-area distributions of $O_2/N_2$ ratio, from 0–14 km and 87° N to 85° S, using measurements from a series of aircraft campaigns such as five HIAPER Pole-to-Pole Observations campaigns (HIPPO) in 2009-2011 (Wofsy et al., 2011), and the $O_2/N_2$ Ratio and $CO_2$ Airborne Southern Ocean (ORCAS) study in 2016 (Stephens et al., 2018).

Stephens et al. (2021) also conducted continuous observations of $O_2$ mole fraction using a vacuum ultraviolet (VUV) absorption detector (Stephens et al., 2003). They adjusted the continuous $O_2$ data to the simultaneously observed flask-based $O_2/N_2$ ratio corrected for the artificial fractionation using $Ar/N_2$ ratio, since the artificial fractionations for the continuous observations were more significant than that for the flask sampling due to the lower flow rate. Based on the continuous $O_2$ data, Morgan et al. (2019) reported summertime vertical gradients of the atmospheric $O_2/N_2$ ratio and $CO_2$ amount fraction through the atmospheric boundary layer over the Drake Passage region of the Southern Ocean, to evaluate the air-sea $O_2/CO_2$ flux ratios in the region. Aircraft observations of $Ar/N_2$ are also used to evaluate gravitational separation of the atmospheric

components, which is an indicator of the Brewer-Dobson circulation (e.g. Ishidoya et al., 2013), in the lowermost stratosphere under the condition that the artificial fractionation is reduced sufficiently (Ishidoya et al., 2008; Birner et al., 2020).

In this study, as an update to Ishidoya et al. (2014), we present 9-year-long $O_2/N_2$ ratio variations observed in the troposphere over the western North Pacific. Measurements were carried out on monthly scheduled cargo aircraft flights with a fixed flight route, and the thermally-diffusive artificial fractionations on $O_2/N_2$ ratio were corrected by using simultaneously-

measured $Ar/N_2$ ratio. Using these corrected values, we made precise evaluation of the altitude-latitude distributions of seasonal cycle, vertical profile and year-to-year variation along the flight route. We mainly focus our discussions on the variations in the tropospheric APO and $CO_2$ amount fraction with the aid of a 3-D atmospheric chemistry-transport model. We also estimate average terrestrial biospheric and oceanic $CO_2$ uptakes for the period 2012–2019, by using long-term trends of $O_2/N_2$ ratio and $CO_2$ amount fraction.

**2 Methods**

The cargo aircraft C-130 flies once per month from Atsugi Base (35.45° N, 139.45° E), Kanagawa, Japan, to Minamitorishima, Japan, a small coral atoll (MNM; 24.29° N, 153.98° E). Two types of C-130, C-130H and C-130R, have been used for the observations. The cruising altitude is about 6 km, and 24 air samples were pressurized into 1.7 L silica-coated titanium flasks to an absolute pressure of 0.4 MPa during the flight. A set of 17–20 samples were collected during the level

flight while others were collected during the descent portion at MNM. Flask air sampling for the C-130H flight is manually operated by two JMA personnel on board the aircraft. Therefore, a diaphragm pump is modified to be operated by hand, without the electric power supply. Air samples are collected from the air-conditioning system in the C-130H aircraft. Fresh air outside the aircraft is compressed by an engine and fed into the air-conditioning ducts by passing it through a pneumatic system and air cycle packs. A Teflon tube with a diameter of 1/4 in. is inserted into the air-conditioning blowing nozzle upstream of

the recirculation fan for the air sampling, so that the sample air is not contaminated with the cabin air. The open end of the air sampling intake is situated at wing root and faces the front side of the aircraft. Unfortunately, details of the air sampling line from the inlet to flask sampler have not been informed to researchers from Japan Ministry of Defense. It is noted that we only use Teflon tube in the upstream of the diaphragm pump to avoid absorption and/or permeation of $O_2$ due to a pressurization of the Teflon tube. In this regard, we found measured values of $O_2/N_2$ ratio for the air samples corrected in January, 2016 were

anomalously low, when Teflon tube was used in the downstream of the diaphragm pump. Therefore, we exclude the $O_2/N_2$ ratio values in January 2016 from the analyses in this study. During other flights, we use a flexible tube made of stainless steel in the downstream of the diaphragm pump. Details of the air sampling method has been described elsewhere (Tsuboi et al., 2013).

The flask air samples were brought back to the Japan Meteorological Agency (JMA) and analyzed for $CO_2$, $CH_4$, $CO$

and $N_2O$ amount fractions. The observational results are reported by Niwa et al. (2014). The $CO_2$ amount fraction was measured using a non-dispersive infrared analyzer (Licor, LI-7000) with a precision of better than ±0.07 μmol mol$^{-1}$ (Tsuboi et al., 2013)

against the World Meteorological Organization (WMO) mole fraction scales for $CO_2$ (Zhao and Tans, 2006). The dataset is posted on the WMO's World Data Centre for Greenhouse Gases (WMO/WGCGG, https://doi.org/10.50849/WDCGG_0001-8002-1001-05-02-9999). After the JMA analyses, the flasks were sent to the National Institute of Advanced Industrial Science and Technology (AIST) to measure $O_2/N_2$ and $Ar/N_2$ ratios, as well as the stable isotopic ratios of $N_2$, $O_2$ and Ar (Ishidoya et al., 2014). In this study, we present the measured data obtained from the air samples collected for the period May 2012 – March 2020. In Fig. 1, we show all the locations where the air samples were collected onboard C-130 aircrafts during the observation period, and the locations of MNM, Atsugi Base and Ioto Island (24.76° N, 141.29° E), Japan.

The values of $\delta(O_2/N_2)$, $\delta(Ar/N_2)$, stable isotopic ratios of $N_2$, $O_2$ and Ar ($\delta(^{15}N)$, $\delta(^{18}O)$ and $\delta(^{40}Ar)$) are reported in per meg (one per meg is equal to $1 \times 10^{-6}$):

$$\delta(O_2/N_2) = \frac{R_{\text{sample}}(^{16}O^{16}O/^{14}N^{14}N) - R_{\text{standard}}(^{16}O^{16}O/^{14}N^{14}N)}{R_{\text{standard}}(^{16}O^{16}O/^{14}N^{14}N)}, \quad (1)$$

$$\delta(Ar/N_2) = \frac{R_{\text{sample}}(^{40}Ar/^{14}N^{14}N) - R_{\text{standard}}(^{40}Ar/^{14}N^{14}N)}{R_{\text{standard}}(^{40}Ar/^{14}N^{14}N)}, \quad (2)$$

$$\delta(^{15}N) = \frac{R_{\text{sample}}(^{15}N^{14}N/^{14}N^{14}N) - R_{\text{standard}}(^{15}N^{14}N/^{14}N^{14}N)}{R_{\text{standard}}(^{15}N^{14}N/^{14}N^{14}N)}, \quad (3)$$

$$\delta(^{18}O) = \frac{R_{\text{sample}}(^{18}O^{16}O/^{16}O^{16}O) - R_{\text{standard}}(^{18}O^{16}O/^{16}O^{16}O)}{R_{\text{standard}}(^{18}O^{16}O/^{16}O^{16}O)}, \quad (4)$$

$$\delta(^{40}Ar) = \frac{R_{\text{sample}}(^{40}Ar/^{36}Ar) - R_{\text{standard}}(^{40}Ar/^{36}Ar)}{R_{\text{standard}}(^{40}Ar/^{36}Ar)}, \quad (5)$$

where the subscripts 'sample' and 'standard' refer to the values of the sample and standard air, respectively. The values of $\delta(O_2/N_2)$, $\delta(Ar/N_2)$, $\delta(^{15}N)$, $\delta(^{18}O)$ and $\delta(^{40}Ar)$ of the air samples were determined against our primary standard air (cylinder No. CRC00045) by using a mass spectrometer (Thermo Scientific Delta-V) (Ishidoya and Murayama, 2014) with a respective reproducibility of about 5, 8, 1.5, 3 and 13 per meg ($1\sigma$). The scale based on the primary standard air is our original scale and called as "EMRI/AIST scale" in Aoki et al. (2021).

As already discussed in Ishidoya et al. (2014), the measured values of $\delta(O_2/N_2)$ are contaminated by significant artificial thermally-diffusive fractionation of $O_2$ and $N_2$ during the air sample collection process onboard the aircraft. Figure 2 shows the relationships between $\delta(Ar/N_2)$, $\delta(^{18}O)$ and $\delta(^{40}Ar)$ with $\delta(^{15}N)$ for all the air samples analyzed in this study. It was found that $\delta(Ar/N_2)$, $\delta(^{18}O)$ and $\delta(^{40}Ar)$ change linearly in proportion to $\delta(^{15}N)$, and the linear regression analyses gave respective slopes of $(16.3 \pm 0.1)$, $(1.58 \pm 0.01)$ and $(2.69 \pm 0.05)$ per meg (per meg)$^{-1}$ for the $\delta(Ar/N_2)/\delta(^{15}N)$, $\delta(^{18}O)/\delta(^{15}N)$ and $\delta(^{40}Ar)/\delta(^{15}N)$ ratios. These ratios agree well with the ratios of $(16.2 \pm 0.1)$, $(1.55 \pm 0.02)$ and $(2.75 \pm 0.05)$ for $\delta(Ar/N_2)/\delta(^{15}N)$, $\delta(^{18}O)/\delta(^{15}N)$ and $\delta(^{40}Ar)/\delta(^{15}N)$, respectively, determined from the laboratory experiments on the effect of thermally-diffusive fractionations on $\delta(Ar/N_2)$, $\delta(^{15}N)$, $\delta(^{18}O)$ and $\delta(^{40}Ar)$ (Ishidoya et al., 2013). Therefore, we decided to correct for the thermally-diffusive fractionation of $O_2$ and $N_2$ on the observed $\delta(O_2/N_2)$ by using the following equation (Ishidoya et al., 2014):

$$\delta_{\text{cor.}}(O_2/N_2) = \delta_{\text{meas.}}(O_2/N_2) - \alpha_{O_2} \cdot \alpha_{Ar}^{-1} \cdot \Delta\delta_{\text{meas.}}(Ar/N_2), \quad (6)$$

Here, $\delta_{cor.}(O_2/N_2)$ and $\delta_{meas.}(O_2/N_2)$ denote the corrected and measured $\delta(O_2/N_2)$, respectively. The coefficients $\alpha_{O2} = (4.57 \pm 0.02)$ and $\alpha_{Ar} = (16.2 \pm 0.1)$ are the $\delta(O_2/N_2)/\delta(^{15}N)$ and $\delta(Ar/N_2)/\delta(^{15}N)$ ratios respectively, determined from the laboratory experiments as described by Ishidoya et al. (2013). The value for $\alpha_{O2}$ is not directly reported in Ishidoya et a. (2013) but it was obtained from the same laboratory experiment (Fig. 2 in their study). The $\alpha_{O2}/\alpha_{Ar}$ ratio of 4.57/16.2 is close to the Keeling et al. (2004) diffusion factor for $(Ar/N_2)/(O_2/N_2)$ and results in the same tracer $\delta(O_2/N_2)^*$ (Stephens et al., 2020). $\Delta\delta_{meas.}(Ar/N_2)$ is the deviation of the measured $\delta(Ar/N_2)$ from its reference point that is determined by using the annual mean value of $\delta(Ar/N_2)$ in 2013 observed at the surface in Tsukuba (36° N, 140° E), Japan (Ishidoya and Murayama, 2014). Therefore, the effect of the seasonal $\delta(Ar/N_2)$ cycle on $\delta_{cor.}(O_2/N_2)$ was not excluded in this study. This could lead to an over and under correction of the surface $\delta_{cor.}(O_2/N_2)$ by about 2 per meg in the summertime and wintertime, respectively. Considering the measurement reproducibility of $\delta_{meas.}(O_2/N_2)$ of 5 per meg, uncertainty of the correction of 2 per meg (8 per meg x $\alpha_{O2} \alpha_{Ar}^{-1}$; 8 per meg is the measurement reproducibility of $\delta_{meas.}(Ar/N_2)$) and the $\delta(Ar/N_2)$ seasonality, the overall uncertainty of $\delta_{cor.}(O_2/N_2)$ was evaluated to be less than 6 per meg. It is noted that the annual mean value of $\delta(Ar/N_2)$ also shows interannual variation with a peak-to-peak amplitude of 9 per meg $a^{-1}$ and secular trend of 0.75 per meg $a^{-1}$ (Ishidoya et al., 2021). They result in uncertainties of $\delta_{cor.}(O_2/N_2)$ by 2.5 and 0.1 per meg $a^{-1}$ for the interannual variation and secular trend, respectively, which are significantly smaller than those of the observed $\delta_{cor.}(O_2/N_2)$ discussed below. The details of the correction of artificial fractionations of $O_2$ and $N_2$ is given in Ishidoya et al. (2014). It is noted that the correction by using $\delta(^{15}N)$, which is stable in the atmosphere over long time period, is also suitable to obtain $\delta_{cor.}(O_2/N_2)$. Since the uncertainty of the corrected $\delta_{cor.}(O_2/N_2)$ by using $\delta(^{15}N)$ is ±7 per meg (±1.5 x 4.57), which is larger than that by using $\delta(Ar/N_2)$ (±2 per meg), we determined to use $\delta(Ar/N_2)$ rather than $\delta(^{15}N)$ in the present study.

We used $\delta(APO)$ for the detailed analyses of the air-sea exchange of $O_2$. $\delta(APO)$ was calculated from the observed $\delta_{cor.}(O_2/N_2)$ and $CO_2$ amount fraction;

$$\delta(APO) = \delta_{cor.}(O_2/N_2) + \frac{\alpha_B}{X(O_2)} y(CO_2) - 2000 \times 10^{-6}, \quad (7)$$

where $y(CO_2)$ is the dry amount fraction of $CO_2$, $\alpha_B$ is the OR of 1.1 for terrestrial biospheric activities, $X(O_2)$ of 0.2093 – 0.2094 is the amount fraction of atmospheric $O_2$ (Tohjima et al., 2005; Aoki et al., 2019), and 2000 is an arbitrary reference. We use 0.2094 for the calculation of $\delta(APO)$. From the definition, $\delta(APO)$ is conserved for terrestrial biospheric activities, and driven by not only air-sea exchange of $O_2$, $N_2$ and $CO_2$ but also fossil fuel consumption of which OR value is generally larger than 1.1 (Keeling, 1988). In order to investigate the observed $\delta_{cor.}(O_2/N_2)$ variations, we used a three-dimensional atmospheric transport model NICAM-TM (Niwa et al., 2011) to simulate $CO_2$ amount fraction and $\delta(APO)$ using surface $O_2$, $N_2$ and $CO_2$ fluxes. NICAM-TM is based on the Nonhydrostatic ICosahedral Atmospheric Model (NICAM: Satoh et al., 2008, 2014) and its tracer transport version has been used for atmospheric transport and flux inversion studies of greenhouse gases (e.g. Niwa et al. 2012). The horizontal model resolution used in this study had a mean grid interval of about 112 km. The

model was driven by nudging the horizontal winds towards the Japanese 55-year Reanalysis data (JRA-55: Kobayashi et al., 2015).

The surface fluxes incorporated into NICAM-TM were the air-sea fluxes of $O_2$, $N_2$ and $CO_2$, and also $CO_2$ and $O_2$ fluxes from fossil fuel combustion. The air-sea $O_2$ and $N_2$ fluxes were the climatological seasonal anomalies taken from the TransCom experimental protocol (Blaine, 2005; Garcia and Keeling 2001). The fluxes were computed to give the seasonal component and the annual mean values at every grid point to be zero. The air-sea $CO_2$ flux was obtained from the monthly sea surface $CO_2$ flux climatology of Takahashi et al. (2009). The CDIAC fossil fuel database was used for the fossil fuel $CO_2$ flux (Andres

et al., 2016; Gilfillan et al., 2019). Model-based changes in $\delta(APO)$, $CO_2$ amount fraction and $\delta(O_2/N_2)$ ($\Delta\delta(APO)$, $\Delta y(CO_2)$ and $\Delta\delta(O_2/N_2)$) were calculated using the following equations (e.g. Nevison et al., 2008; Tohjima et al., 2012) in per meg, $\mu$mol mol$^{-1}$ and per meg, respectively:

$$\Delta\delta(APO) = \left(\frac{\Delta y^{SA}(O_2)}{X(O_2)} - \frac{\Delta y^{SA}(N_2)}{X(N_2)}\right) + \left(\frac{-\alpha_F \cdot \Delta y^{FF}(CO_2) + \alpha_B \cdot \Delta y^{FF}(CO_2)}{X(O_2)}\right) + \frac{\alpha_B \cdot \Delta y^{OC}(CO_2)}{X(O_2)}, \quad (8)$$

$$\Delta y(CO_2) = \Delta y^{FF}(CO_2) + \Delta y^{OC}(CO_2) + \Delta y^{TB}(CO_2), \quad (9)$$

$$\Delta\delta(O_2/N_2) = \left(\frac{\Delta y^{SA}(O_2)}{X(O_2)} - \frac{\Delta y^{SA}(N_2)}{X(N_2)}\right) + \frac{-\alpha_F \cdot \Delta y^{FF}(CO_2)}{X(O_2)} + \frac{-\alpha_B \cdot \Delta y^{TB}(CO_2)}{X(O_2)}, \quad (10)$$

where $\Delta y(O_2)$, $\Delta y(N_2)$ and $\Delta y(CO_2)$ are changes in dry amount fractions of the respective gases calculated using NICAM-TM. The superscripts "SA", "FF", "OC" and "TB" denote the seasonal anomaly of the air-seas $O_2$ and $N_2$ flux, $CO_2$ flux from fossil fuel combustion, ocean and terrestrial biosphere, respectively. $X(O_2)$ and $\alpha_B$ have the same meaning as in Equation (7), while $X(N_2)$ is the dry amount fraction of $N_2$ in the atmosphere and $\alpha_F$ is the global average OR for fossil fuel combustion. In this

study, we adopted $X(N_2) = 0.7808$ and $\alpha_F = 1.37$. The $\alpha_F$ was calculated from the fossil fuel and cement production emissions by fuel type for the period 2012-2019, reported by GCP (Friedlingstein et al., 2020), and the oxidative ratios for the different fuel type were taken from Keeling (1988). It should be noted that we assume initial amount fractions of $y(O_2)$ and $y(N_2)$ are equal to $X(O_2)$ and $X(N_2)$, respectively, in eqs. (8) and (10). We can rewrite equation (8) as:

$$\Delta\delta(APO) = \Delta\delta^{SA}(APO) + \Delta\delta^{FF}(APO) + \Delta\delta^{OC}(APO). \quad (11)$$

Finally, $\delta(O_2/N_2)$, $y(CO_2)$ and $\delta(APO)$ obtained from NICAM-TM are reported as respective deviations of $\Delta\delta(APO)$, $\Delta y(CO_2)$ and $\Delta\delta(O_2/N_2)$ from arbitrary reference points, in other words, reported on different scales from those of observations. The $\delta(APO)$ simulation run incorporating the above-mentioned surface fluxes is referred to as the "control run". In this calculation, $\delta(APO)$ driven by an annual mean air-sea $O_2$ and $N_2$ fluxes (hereafter referred to as the "$\delta^{AM}(APO)$"), which were estimated by Gruber et al. (2001), was ignored. In other words, the $\delta(APO)$ obtained from the NICAM-TM control run ignore the

contribution of not only interannual variations but also spatial distribution in annual mean air-sea $O_2$ and $N_2$ fluxes. If the control run represents other components other than $\delta^{AM}(APO)$ correctly, then the contribution of $\delta^{AM}(APO)$ was evaluated by subtracting the simulated $\delta(APO)$ from the observed $\delta(APO)$. In Sections 3-2, we will discuss interannual variation in this context.

## 3 Results and discussion

### 3.1 Latitudinal and vertical distributions of $\delta_{cor.}(O_2/N_2)$, CO$_2$ amount fraction and $\delta$(APO)

Figure 3(a) shows the measured $\delta(O_2/N_2)$ and $\delta(Ar/N_2)$ for all the air samples observed in this study, and Fig. 3(b) shows the $\delta_{cor.}(O_2/N_2)$ corrected values by applying eq. (6) to the measured values. As seen from the figures, significant artificial fractionations found in the measured $\delta(O_2/N_2)$ are reduced dramatically by the correction. It can also be seen from Fig. 3(a) that the fractionations have become smaller since 2018, especially at the higher altitude. It should be noted that these noted changes across 2018 could be at least partly related to changes in the aircraft type from C-130H to C-130R used for flask sampling. The periods when the C-130R were used are indicated by pale red shade. No systematic data gaps were found in the $\delta_{cor.}(O_2/N_2)$ time series across 2018, so that we successfully corrected the fractionation of O$_2$ and N$_2$ both for C-130H and C-130R. To examine the validity of the significant corrections applied to the $\delta_{meas.}(O_2/N_2)$, we also show the vertical profiles of detrended $\delta_{cor.}(O_2/N_2)$ separated by season in Fig. 3(c), obtained by subtracting a linear secular trend from the data in Fig. 3(b). As seen from the figure, seasonal $\delta_{cor.}(O_2/N_2)$ variations with summertime maxima are clearly seen at all altitudes, which supports validity of the correction. We also presented some examples that synoptic-scale variations in $\delta(O_2/N_2)$ can also be reproduced by using $\delta_{cor.}(O_2/N_2)$ in our previous study (Fig. 4 in Ishidoya et al. (2014)). Since details of the air sampling line from the inlet to flask sampler have not been informed to researchers, it is difficult to explain the cause(s) of the much lower variability in $\delta(Ar/N_2)$ data for periods of C-130R. The visible difference is the location of the air-conditioning blowing nozzles of C-130H and C-130R. The former and the latter are attached at the roof in the cockpit and at the front of the assistant driver's seat, respectively, and air sampling tubes are inserted into the nozzles.

Figure 4(a) shows variations in the $\delta_{cor.}(O_2/N_2)$, CO$_2$ amount fraction and $\delta$(APO) observed in the layer $(6.1 \pm 0.5)$ km $(\pm 1\sigma)$ (hereafter referred to as "middle troposphere") at five latitudes over the western North Pacific. Best-fitted curves to the data and secular trends obtained using a digital filtering technique (Nakazawa et al., 1997a) are also shown. Using this filtering technique, the average seasonal cycles were approximated by the sum of the fundamental and its first harmonic with the respective periods of 12 and 6 months. The residuals obtained by subtracting the approximated average seasonal cycle from the data were interpolated linearly to calculate daily values of $\delta_{cor.}(O_2/N_2)$, CO$_2$ amount fraction and $\delta$(APO). The daily values were smoothed by the 26th-order Butterworth filter with a cutoff period of 36 months to derive the long-term trend. The long-term trend thus obtained was subtracted from the data, and the average seasonal cycle was determined again from the residuals. These steps were repeated until an unchangeable long-term trend obtained. The observational data deviated from the best-fitted curves more than $\pm 3\sigma$ are excluded from the analyses and not shown in the figures discussed in this study (2% of the observational data are excluded in total). As can be seen in Fig. 4(a), secular decreases in $\delta_{cor.}(O_2/N_2)$ and $\delta$(APO) and increases in CO$_2$ amount fraction, accompanied by prominent seasonal cycles, were observed at each latitude. The secular changes in $\delta_{cor.}(O_2/N_2)$ and CO$_2$ amount fraction can be attributed mainly to O$_2$ consumption and CO$_2$ emission resulting from fossil fuel combustion. The seasonally dependent air-sea O$_2$ flux and the terrestrial biospheric activity contribute towards the observed

seasonal $\delta(O_2/N_2)$ cycle, while the terrestrial biospheric activity is the main contributor to the seasonal $CO_2$ amount fraction cycle (e.g. Keeling et al., 1993; Keeling and Manning, 2014). The average rates of change in the observed $\delta_{cor.}(O_2/N_2)$, $CO_2$ amount fraction and $\delta(APO)$ at the four latitudes shown in Fig. 4(a) were $(-24.2 \pm 0.4)$ per meg $a^{-1}$, $(2.43 \pm 0.05)$ μmol mol$^{-1}$ $a^{-1}$ and $(-11.3 \pm 0.4)$ per meg $a^{-1}$, respectively, for the observational period. General features of the observed variations in

$\delta(O_2/N_2)$, $CO_2$ amount fraction and $\delta(APO)$ are well reproduced by the control run of NICAM-TM, as shown in Fig. 4(b). Figure 5(a) shows variations in $\delta_{cor.}(O_2/N_2)$, $CO_2$ amount fraction and $\delta(APO)$ observed over MNM. Clear secular trends and prominent seasonal cycles of $\delta_{cor.}(O_2/N_2)$, $CO_2$ amount fraction and $\delta(APO)$ are distinguishable, similar to those in Fig. 4(a). We also note in Fig. 5(a) that the seasonal amplitudes of $\delta_{cor.}(O_2/N_2)$, $CO_2$ amount fraction and $\delta(APO)$ decrease with increasing altitudes. These features are also reproduced by the control run of NICAM-TM (Fig. 5(b)).

Figure 6(a) shows average seasonal cycles of $\delta(APO)$ and $CO_2$ amount fraction observed at five latitudes over the western North Pacific. They show clear seasonal cycles with summertime maxima in $\delta(APO)$ and minima in $CO_2$. However, the amplitude of seasonal $\delta(APO)$ cycle decreases significantly toward the lower latitudes, with seasonal maxima and minima clearly occurring earlier than those of the $CO_2$ cycle. Figure 6(b) shows the corresponding average seasonal cycles of $\delta(APO)$ and $CO_2$ amount fraction obtained from the control run of NICAM-TM. Earlier appearances of the seasonal maxima and

minima in $\delta(APO)$ than those in $CO_2$ amount fraction are reproduced by NICAM-TM. The seasonal amplitude of the simulated $\delta(APO)$ also decreases toward the lower latitudes, in agreement with the observation, but is underestimated. As for the $CO_2$ amount fraction, the simulated seasonal cycles agree well with the observations. In Figs. 6(c) and (d), we also show same average seasonal cycles of the observed and simulated $\delta(APO)$ in Figs. 6(a) and (b), respectively, along with the detrended observed values at each latitude for the convenience of visual comparison between the observed and simulated results.

Figure 7(a) shows average seasonal cycles of $\delta(APO)$ and $CO_2$ amount fraction observed over MNM. The $\delta(APO)$ seasonal cycle varies in opposite phase from the $CO_2$ amount fraction. However, amplitudes of the seasonal $\delta(APO)$ cycle decrease significantly with the higher altitude, with seasonal minima clearly occurring earlier than those of the $CO_2$ cycle. These salient characteristics are reproduced generally by the NICAM-TM control run (Fig. 7(b)). Similar to Fig. 6, we also show Figs. 7(c) and (d) for visual comparison between the observed and simulated results.

In order to identify and explore some of the cause(s) that gave rise to the observed differences in the latitudinal/altitudinal changes in the seasonal cycles between $\delta(APO)$ and $CO_2$ amount fraction shown in Fig. 6(a) and 7(a), we carried out additional NICAM-TM simulations. In the calculation, we used the same fluxes that were used in the control run but for the northern hemispheric flux only for the TransCom seasonal climatology (hereafter referred to as "without-SH-flux run"). It is well known that the opposing phase of the seasonal $\delta(APO)$ cycles between the northern and southern hemispheres is due to seasonal

changes in the air-sea $O_2$ ($N_2$) flux with summertime maxima (e.g. Keeling et al., 1998; Tohjima et al., 2012). Therefore, in comparing the control run with the without-SH-flux run, we decided to evaluate changes in the seasonal $\delta(APO)$ cycle by superimposing the anti-phase seasonal cycles through the inter-hemispheric mixing of air. On the other hand, seasonal $CO_2$

amount fraction cycle is much smaller in the southern hemisphere than that in the northern hemisphere (e.g. Nakazawa et al., 1997b). Therefore, it is expected that the seasonal cycle in $CO_2$ amount fraction in the northern hemisphere does not change significantly by the inter-hemispheric atmospheric mixing.

The seasonal $\delta(APO)$ cycles obtained from the without-SH-flux run are also shown in Figs. 6(b) and 7(b). As seen from Fig. 6(b), latitudinal differences in the seasonal $\delta(APO)$ amplitudes from the without-SH-flux run are clearly smaller than those from the control run. In addition, appearances of the maxima and minima of the seasonal $\delta(APO)$ cycle from the without-SH-flux run are later than those from the control run, pushing the timing closer to those of seasonal $CO_2$ amount fraction cycles. The seasonal $\delta(APO)$ cycles from the without-SH-flux run in Fig. 7(b) also show similar features. These results suggest that the inter-hemispheric mixing of air modifies the seasonal $\delta(APO)$ cycles significantly, especially in the higher altitude in the lower latitude region. To compare, in more detail, the observed latitudinal/altitudinal variation in the seasonal $\delta(APO)$ cycle amplitude with those calculated by NICAM-TM, Fig. 8(a) shows a latitudinal distribution of average fractions of the observed seasonal $\delta(APO)$ and $CO_2$ amount fraction amplitudes relative to the 25.5° N values. The decrease in the seasonal amplitude of $\delta(APO)$ toward the lower latitude is about 50%, while that of $CO_2$ amount fraction is less than 10%; these are well reproduced by the control run of NICAM-TM. On the other hand, the without-SH-flux run yields a decrease in the $\delta(APO)$ amplitude by 20% toward the lower latitude, which is significantly smaller than that from the $\delta(APO)$ from control run, but slightly larger than that of $CO_2$ amount fraction. These results indicate that the SH makes a significant contribution to the amplitude and phase of the seasonal $\delta(APO)$ cycle at lower latitude.

Figure 8(b) shows average fractions of the seasonal amplitudes of $\delta(APO)$ and $CO_2$ amount fraction with altitude, relative to the corresponding values at 1.3 km. The average fractions of surface seasonal cycles, obtained from continuous observations of $\delta(O_2/N_2)$ and $CO_2$ amount fraction at MNM since December 2015 (updated from Ishidoya et al., 2017), are also plotted. The seasonal amplitude of $\delta(APO)$ decreases rapidly with altitude by about 70%, while that of $CO_2$ amount fraction decreases by less than 20%. The altitudinal dependence of the seasonal $\delta(APO)$ amplitude is not consistent with that of $CO_2$. This could be attributed to the fact that the seasonal $CO_2$ cycle is driven mainly by terrestrial biospheric activities in the northern mid- and high latitudes and the seasonal minimum appears in summer, which makes the seasonal $CO_2$ cycle uniform from the surface to the middle troposphere due to continental strong convection in summer. On the other hand, seasonal APO cycle is driven by air-sea flux, so that altitudinal homogenization of the seasonal APO cycle due to convection is weaker than that of $CO_2$. These features are well reproduced by the control run of NICAM-TM. The altitudinal decrease in the seasonal $\delta(APO)$ amplitude is also reproduced by the without-SH-flux run, although a slight underestimation is found above 5 km. Therefore, the altitudinal decrease in the seasonal $\delta(APO)$ amplitude over MNM is mainly due to an attenuation of the seasonal air-sea $O_2$ and $N_2$ fluxes around MNM with altitude, with some influence from the inter-hemispheric atmospheric mixing. Consequently, over the western North Pacific region, the altitude-latitude distribution of seasonal $\delta(APO)$ cycles is more sensitive to the atmospheric transport processes associated with inter-hemispheric air mixing and vertical attenuation of surface signal, compared with those of seasonal $CO_2$ amount fraction cycles.

We also compared the observed and simulated annual mean values of $\delta(APO)$ and $CO_2$ amount fraction. Figure 9(a) shows average deviations of the middle tropospheric annual mean values of $\delta(APO)$ and $CO_2$ amount fraction at each latitude, relative to the 25.5° N values. The deviation values of $\delta(APO)$ and $CO_2$ amount fraction are well reproduced by the control and without-SH-flux runs. Therefore, the surface fluxes of $O_2$, $N_2$ and $CO_2$ in the northern hemisphere are the main contributors to the observed latitudinal variations in Fig. 9(a). As discussed in connection with eq. (11), we ignored $\delta^{AM}(APO)$ in our $\delta(APO)$ simulation using NICAM-TM, which is a component of $\delta(APO)$ driven by annual mean air-sea $O_2$ and $N_2$ fluxes. Therefore, the results of our simulation suggest that $\delta^{AM}(APO)$ does not affect significantly the latitudinal variations in the annual mean values of the middle tropospheric $\delta(APO)$ at 25-34° N. Figure 9(b) shows the altitude deviations of the annual mean values of $\delta(APO)$ and $CO_2$ amount fraction, relative to their corresponding values at 1.3 km over MNM. The observed profile of $CO_2$ amount fraction is well reproduced by NICAM-TM. On the other hand, the average vertical gradient of $\delta(APO)$ profiles, obtained from the control and without-SH-flux runs of NICAM-TM, seems to be slightly larger than the observation. This may be due to the ignored contribution of $\delta^{AM}(APO)$; in that case, it may be that the sea area around MNM emits $O_2$ to the atmosphere throughout the observation period. Moreover, it is clearly seen from the figure that the interannual variation in the observed $\delta(APO)$ profiles is much larger than that in the NICAM-TM simulations. This may also be due to the ignored contribution of $\delta^{AM}(APO)$, and we discuss interannual variations in the observed and simulated $\delta(APO)$ in the section below.

## 3.2 Interannual variations in $\delta(APO)$ and its implication to global air-sea $O_2$ flux and $CO_2$ budget

In this section, we discuss causes of the interannual variations found in the middle tropospheric $\delta(APO)$ observed over the western North Pacific. Figure 10(a) shows same secular trends in Fig. 4(a) and their annual change rates along with the deseasonalised values of $\delta(APO)$ at each latitude. The change rates observed at 29-34° N show interannual variation with maxima around the early 2015 and mid 2019, with a minimum at all latitudes in the early 2017. The corresponding change rates obtained from the control run of NICAM-TM are also shown in Fig. 10(b). The change rates obtained from NICAM-TM at 29-34° N show interannual variations in phase with the observed rates, although the amplitudes are smaller by about 80%. The interannual variations observed at 24-28° N are also larger than the simulated values. Therefore, it is likely that interannual variation in the $\delta^{AM}(APO)$, which was not incorporated into NICAM-TM, is a main contributor to the observed interannual variations at various latitudes. In this connection, it is possible that the interannual variation in the global air-sea $CO_2$ flux could also contribute to the larger interannual variation in the observed $\delta(APO)$ since the NICAM-TM model incorporated only the monthly sea surface $CO_2$ flux climatology to calculate $\delta^{OC}(APO)$. However, the global air-sea $CO_2$ flux reported by the Global Carbon Project (GCP) (Friedlingstein et al., 2020) showed an interannual variation of 0.07 Pg $a^{-1}$ during 2012-2019, corresponding to 0.2 per meg $a^{-1}$ of $\delta^{OC}(APO)$, which is much smaller than the interannual variation in the observed $\delta(APO)$ shown in Fig. 10.

By assuming that all other components besides $\delta^{AM}$(APO) are well represented in the NICAM-TM control run, we subtracted the change rates simulated by NICAM-TM from the observed rates, to extract the interannual variations due only to the $\delta^{AM}$(APO). The calculated change rates of $\delta^{AM}$(APO) are shown at the bottom of Fig. 10(b). The change rates show similar interannual variations to the observed rates, but the latitudinal differences are smaller. This suggests that the interannual variations driven by $\delta^{AM}$(APO) do not differ significantly as a function of latitude. In the following discussion, we make a bold assumption that an average of the change rates of $\delta^{AM}$(APO) shown in Fig. 10(b) is a global average.

An anomaly of the average interannual variation of $\delta^{AM}$(APO) change rate is shown in Fig. 11 (black line). In this figure, we also plotted a similar anomaly of interannual variation of the $\delta$(APO) change rate due to solubility change (red line, hereafter referred to as "$\delta_{therm}$(APO)"). The $\delta_{therm}$(APO) was calculated from the $\delta(Ar/N_2)$ measurements observed at Tsukuba (36° N, 140° E), Japan (Ishidoya et al., 2021), by multiplying a coefficient of about 0.9 derived from differences in the solubility in $O_2$ and Ar (Weiss 1970). It should be noted that this is a rough approximation of the $\delta_{therm}$(APO), which is a combination of air-sea fluxes of $O_2$, $N_2$, and $CO_2$ caused by solubility changes. As discussed in Ishidoya et al. (2021), the interannual variation in the $\delta(Ar/N_2)$ change rate is in phase with the global ocean heat content reported by ocean temperature measurements (e.g. Levitus et al., 2012). This suggests that $\delta_{therm}$(APO) is also driven by changes in the solubility of the global seawater. By subtracting $\delta_{therm}$(APO) from $\delta^{AM}$(APO), we estimated interannual variation of the $\delta$(APO) change rate due to marine biological activities (green line, hereafter referred to as "$\delta_{netbio}$(APO)"). It is expected that $\delta_{netbio}$(APO) is driven by marine biological activities, not only in the surface mixed layer but also through a ventilation of subsurface low-$O_2$ waters.

Both the $\delta_{therm}$(APO) and $\delta_{netbio}$(APO) show significant interannual variations, roughly in opposite phase with each other. Moreover, the change rate of $\delta_{netbio}$(APO) varies in opposite phase with the NINO.WEST (Japan Meteorological Agency, https://www.data.jma.go.jp/gmd/cpd/db/elnino/index/ninowidx.html), which is an index of El Niño-Southern Oscillation (ENSO). The ENSO is in the El Niño and La Niña phase, respectively, during the period with the negative and positive NINO.WEST index. Therefore, the $\delta_{netbio}$(APO) tends to increase and decrease during El Niño and La Niña, respectively. This is consistent with Eddebber et al. (2017) who examined global and tropical air-sea $O_2$ flux responses to ENSO, based on the Community Earth System Model (CESM). They reported that the upper ocean loses $O_2$ to the atmosphere during El Niño and gains $O_2$ during La Niña mainly due to changes in ventilation of low-$O_2$ waters in the tropical Pacific, the region that has a dominant influence over the interannual variation in global air-sea $O_2$ flux (McKinley et al., 2003). By assuming the interannual variation in the $\delta_{netbio}$(APO) represents a global average, and assuming a 1-box atmosphere with 5.124 x $10^{21}$ g for the total mass of dry air (Trenberth, 1981), 28.97 g mol$^{-1}$ for the mean molecular weight of dry air, and respective fractions of 0.2093 and 0.7808 for $O_2$ and $N_2$ in the atmosphere, we estimated an interannual variation in the global air-sea $O_2$ flux due to marine biological activities (right axis of the green line in Fig. 11). The peak-to-peak amplitude of the $O_2$ flux is found to be about 300 Tmol a$^{-1}$, which is almost consistent with that of global APO flux estimated using the $\delta$(APO) data from Scripps stations (Keeling and Manning, 2014) and a global atmospheric transport inversion (Rödenbeck et al., 2008; Eddebber et al., 2017). Therefore, the seasonal cycles and interannual variations obtained from $\delta_{cor.}(O_2/N_2)$ have a coherent signal we can explain,

although we recognize the number of simplifying assumptions is extensive to derive $\delta_{netbio}$(APO) and the aircraft observations are from a comparatively small region. This helps to prove that the $\delta_{cor.}$(O$_2$/N$_2$) are of good quality, even though the significant correction for enormous sampling artifacts.

By assuming the average secular trends of the middle tropospheric $\delta_{cor.}$(O$_2$/N$_2$) and CO$_2$ amount fraction observed in this study to be global average values, we were able to estimate the global CO$_2$ budget. The equations for separating out the

355 global net terrestrial biospheric and oceanic CO$_2$ uptake are given by Keeling and Shretz (1992) firstly as;

$$B = \frac{\alpha_F}{\alpha_B}F + \frac{1}{0.471}\frac{X(O_2)}{\alpha_B}\frac{d\delta(O_2/N_2)}{dt} - \frac{Z_{eff}}{\alpha_B}, \tag{12}$$

and

$$O = -\frac{1}{0.471}\frac{d}{dt}\left(y(CO_2) + \frac{X(O_2)}{\alpha_B}\delta(O_2/N_2)\right) + \frac{\alpha_B - \alpha_F}{\alpha_B}F + \frac{Z_{eff}}{\alpha_B}. \tag{13}$$

Here, $B$, $F$ and $O$ (in Pg a$^{-1}$, C equivalents) are the global terrestrial biospheric CO$_2$ uptake, the anthropogenic CO$_2$ emitted from fossil fuel combustion and cement manufacturing, and the oceanic CO$_2$ exchange, respectively; d$\delta$(O$_2$/N$_2$)dt$^{-1}$ (per meg a$^{-1}$) and d$y$(CO$_2$)dt$^{-1}$ (μmol mol$^{-1}$ a$^{-1}$) are the observed change rates in atmospheric $\delta_{cor.}$(O$_2$/N$_2$) and CO$_2$ amount fraction, respectively; 0.471 converts CO$_2$ emissions of 1 Pg (C equivalents) to μmol mol$^{-1}$ of atmospheric CO$_2$; $X(O_2)$ is the standard mole fraction of O$_2$ in air (0.2094), $\alpha_F$ and $\alpha_B$ are the OR for global average fossil fuel combustion and net terrestrial biospheric

activities, respectively; and $Z_{eff}$ (Pg yr$^{-1}$) represents the net effect of oceanic O$_2$ outgassing on the oceanic and terrestrial biospheric CO$_2$ uptakes, which has been considered since Bender et al. (2005) and Manning and Keeling (2006). As described in Method section, we use $\alpha_F$ of 1.37 calculated from the fossil fuel and cement production emissions by fuel type reported by GCP.

Long-term change in $Z_{eff}$ is caused mainly by stratification of the ocean and the decrease of O$_2$ solubility in seawater due

to a secular increase in the global ocean heat content (e.g., Bopp et al., 2002). However, as discussed above for Fig. 11, the ocean O$_2$ outgassing shows significant interannual variation, which makes it difficult to estimate year-to-year variations in the global CO$_2$ budget from eqs. (12) and (13). In this regard, Tohjima et al. (2019) estimated the terrestrial biospheric and the oceanic CO$_2$ uptakes using their $\delta$(O$_2$/N$_2$) and CO$_2$ amount fraction data, by changing the time period to obtain average secular change rates. They reported that the CO$_2$ uptakes estimated by using the change rates averaged over a longer period greater

than 5 years were consistent with those reported by GCP (Le Quéré et al., 2018) within $\pm0.5$ Pg a$^{-1}$, while those estimated using annual change rates scattered significantly.

It seems that the averaging period needed to reduce the interannual variations in $\delta_{therm}$(APO) and $\delta_{netbio}$(APO) in Fig. 11, is about 4-5 years, similar to Nevison et al. (2008) and Tohjima et al. (2019). Therefore, we estimated average terrestrial biospheric and oceanic CO$_2$ uptake throughout the observation period (2012-2019; 8-year) to reduce the interannual variation

in $Z_{eff}$ sufficiently. By using the global ocean (0-2000 m) heat content data from the National Oceanographic Data Center (NOAA)/National Centers for Environmental Information (NCEI) (updated from Levitus et al. 2012,

https://www.nodc.noaa.gov/OC5/3M_HEAT_CONTENT/) and the same ratio of air-sea $O_2$ ($N_2$) flux to air-sea heat flux used in Manning and Keeling (2006), we adopted $(0.6 \pm 0.6)$ Pg a$^{-1}$ for $Z_{eff}$ for the period 2012-2019. We assumed 100% uncertainty for $Z_{eff}$ following Manning and Keeling (2006). The value of $F$, $(9.7 \pm 0.5)$ Pg a$^{-1}$, was obtained from emissions from fossil

fuel combustion and industrial processes by GCP (Friedlingstein et al., 2020). By using the average secular trends of $\delta_{cor.}(O_2/N_2)$ and $CO_2$ amount fraction for the observational period of the present study, the respective terrestrial biospheric and oceanic $CO_2$ uptakes were estimated to be $(1.8 \pm 0.9)$ and $(2.8 \pm 0.6)$ Pg a$^{-1}$ for the period 2012-2019. These values agree well with the corresponding $CO_2$ uptake of $(1.8 \pm 1.1)$ and $(2.6 \pm 0.5)$ Pg a$^{-1}$ reported by the GCP (Friedlingstein et al. 2020). It is noted that the terrestrial biospheric $CO_2$ uptake by GCP is calculated by subtracting the $CO_2$ emission due to land-use change

$((1.6 \pm 0.7)$ Pg a$^{-1})$ from their estimated total land $CO_2$ uptake $((3.4 \pm 0.9)$ Pg a$^{-1})$.

## 4 Summary

Regular air samples were taken on cargo aircraft C-130 flights from Atsugi Base to MNM, and air samples have been collected during the level flight and during the descent portion at MNM. In this paper, we have presented the analytical results of the air samples for $CO_2$ amount fraction, $\delta(O_2/N_2)$, $\delta(Ar/N_2)$, $\delta(^{15}N)$ of $N_2$, $\delta(^{18}O)$ of $O_2$ and $\delta(^{40}Ar)$ for the period May 2012 – March

2020. The relationships of $\delta(Ar/N_2)$, $\delta(^{18}O)$ and $\delta(^{40}Ar)$ with $\delta(^{15}N)$ indicate a significant artificial fractionation due to thermal diffusion during the air sample collection.

        The $\delta_{cor.}(O_2/N_2)$ values, corrected for the artificial fractionation by using $\delta(Ar/N_2)$, and the $\delta(APO)$ values derived from $\delta_{cor.}(O_2/N_2)$ were shown to have clear seasonal cycles nearly in opposite phase to that of the $CO_2$ amount fraction from the surface to 6 km along the latitudinal path from 25.5 to 33.5° N. We then used a three-dimensional atmospheric transport

model NICAM-TM that was driven by the air-sea fluxes of $O_2$, $N_2$ and $CO_2$, along with fluxes of $CO_2$ and $O_2$ from fossil fuel combustion, to interpret some of the characteristic features we observed in the seasonal cycles and vertical profiles of APO and $CO_2$ amount fraction.

        Seasonal amplitudes of $\delta(APO)$ and $CO_2$ amount fraction decreased toward the lower latitude from 34.5 to 24.5° N by about 50% and less than 10%, respectively: these features were reproduced by the corresponding ratios from the control run

of NICAM-TM. On the other hand, the without-SH-flux run underestimated the latitudinal change in the $\delta(APO)$ amplitude, which indicated that the seasonal cycle of the mid-tropospheric $\delta(APO)$ was modified significantly by a combination of the northern and southern hemispheric seasonal cycles through the inter-hemispheric atmospheric mixing. The decrease in the $\delta(APO)$ seasonal amplitude was about 70% with altitude from the surface to 6 km, while that of $CO_2$ amount fraction was less than 20%. These features were also reproduced well by the control run of NICAM-TM.

The observed decrease in the annual mean values of $CO_2$ amount fraction with altitude was reproduced by the control run of NICAM-TM. On the other hand, the average vertical gradient of the $\delta(APO)$ profiles was slightly overestimated by the

NICAM-TM simulations, while the simulated interannual variation was underestimated. This may be due to the ignored contribution of $\delta^{AM}(APO)$, which is a component of $\delta(APO)$ driven by annual mean air-sea $O_2$ and $N_2$ fluxes.

The interannual variations in the middle tropospheric $\delta^{AM}(APO)$ were estimated by subtracting the simulated $\delta(APO)$ by the NICAM-TM control run from the observed $\delta(APO)$. We also estimated the solubility-driven component of $\delta(APO)$ ($\delta_{therm}(APO)$) from the $\delta(Ar/N_2)$ observed at Tsukuba, assuming its interannual variation was driven by changes in the globally-averaged solubility of the seawater. The interannual variation in $\delta(APO)$ driven by marine biological activities ($\delta_{netbio}(APO)$) was calculated by subtracting $\delta_{therm}(APO)$ from $\delta^{AM}(APO)$. The $\delta_{netbio}(APO)$ showed significant interannual variations in the opposite phase to that of $\delta_{therm}(APO)$, and the change rate varied in opposite phase with the NINO.WEST. Therefore, the

$\delta_{netbio}(APO)$ values obtained in this study tended to increase and decrease with El Niño and La Niña, respectively, and is in agreement with Eddebber et al. (2017) who examined responses of the global and tropical air-sea $O_2$ flux to ENSO based on CESM.

By assuming the observed secular trends of the middle tropospheric $\delta_{cor.}(O_2/N_2)$ and $CO_2$ amount fraction to be representative of global average values, we estimated terrestrial biospheric and oceanic $CO_2$ uptakes to be $(1.8 \pm 0.9)$ and $(2.8$

$\pm 0.6)$ Pg $a^{-1}$, respectively, for the period 2012-2019. These values agree well with the corresponding $CO_2$ uptake values of $(1.8 \pm 1.1)$ (land) and $(2.6 \pm 0.5)$ Pg $a^{-1}$ (ocean) reported by the GCP.

Additionally, our study has shown that our aircraft observation and the method we used to correct artificial fractionation of $O_2$ and $N_2$ are useful in evaluating inter-hemispheric air mixing processes based on the seasonal $\delta(APO)$ cycle, as well as interannual variations in the global air-sea $O_2$ flux, and in calculating global $CO_2$ budgets based on the long-term trends of

$\delta_{cor.}(O_2/N_2)$ and $CO_2$ amount fraction.

*Data availability.*

The observational data of $\delta_{cor.}(O_2/N_2)$ and $CO_2$ amount fraction are available through the World Data Centre for Greenhouse Gases (WDCGG) at https://gaw.kishou.go.jp, and the respective DOIs are https://doi.org/10.50849/WDCGG_0006-8002-

7001-05-02-9999 and https://doi.org/10.50849/WDCGG_0001-8002-1001-05-02-9999.

*Author contributions.*

SI designed the study, carried out measurements of $\delta(O_2/N_2)$, $\delta(Ar/N_2)$, $\delta(^{15}N)$, $\delta(^{18}O)$ and $\delta(^{40}Ar)$ and drafted the manuscript. KT, SK managed the collections, YN carried out the simulations of NICAM-TM using the supercomputer system (NEC SX-

Aurora TSUBASA) of the National Institute for Environmental Studies (NIES). HM, SM and KI examined the results and provide feedback on the manuscript. All the authors approved the final manuscript.

*Competing interests.*

The authors declare that they have no conflict of interest.

**Acknowledgements.**

The authors gratefully acknowledge many staff members of the Japan Ministry of Defense and Japan Meteorological Agency for supporting the long-term observation.

**Financial support.**

This study was partly supported by the JSPS KAKENHI (grant nos. 15H02814 and 19H01975) and the Global Environment Research Coordination System from the Ministry of the Environment, Japan (grant nos. METI1454 and METI1953).

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

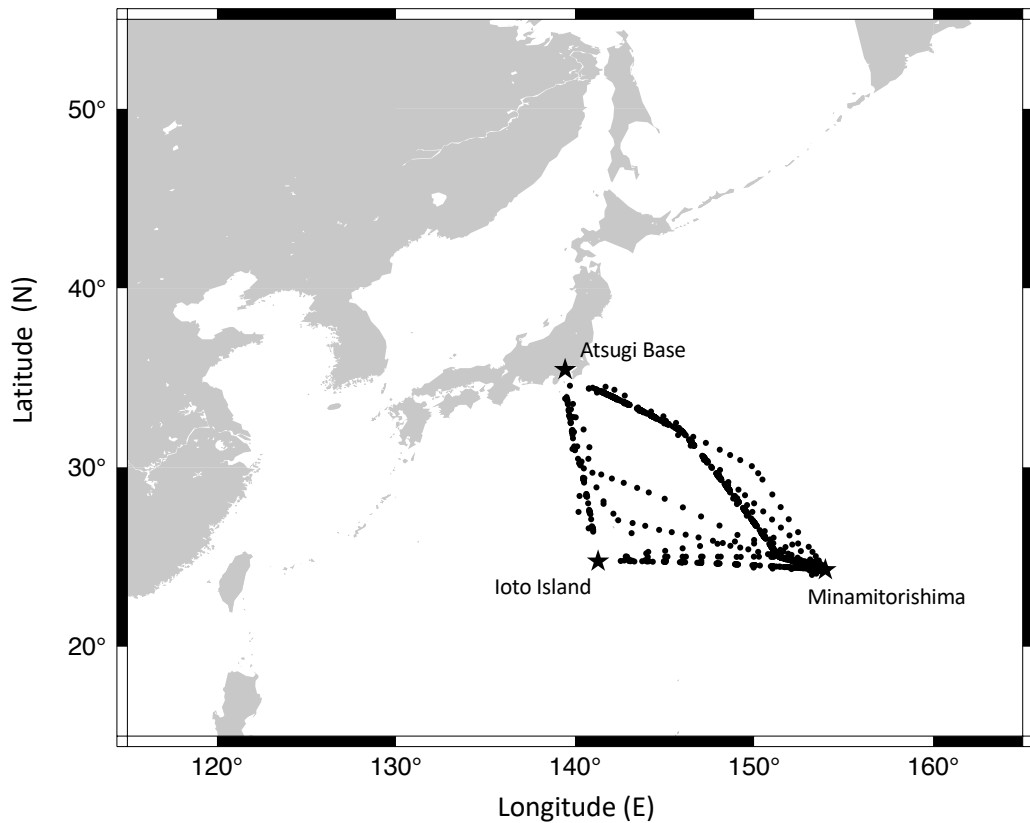

**Figure 1: Locations of C-130 aircraft air sampling for the period May 2012 – March 2020 (circles). Locations of Minamitorishima (MNM), Ioto Island, and Atsugi Base are also shown by stars. Latitudinal and vertical distributions are taken during the level flight and descent portion at MNM, respectively. 2,206 of air samples were collected in total, and 1,783 of them were analyzed for O$_2$/N$_2$ and Ar/N$_2$ ratios, as well as the stable isotopic ratios of N$_2$, O$_2$ and Ar.**

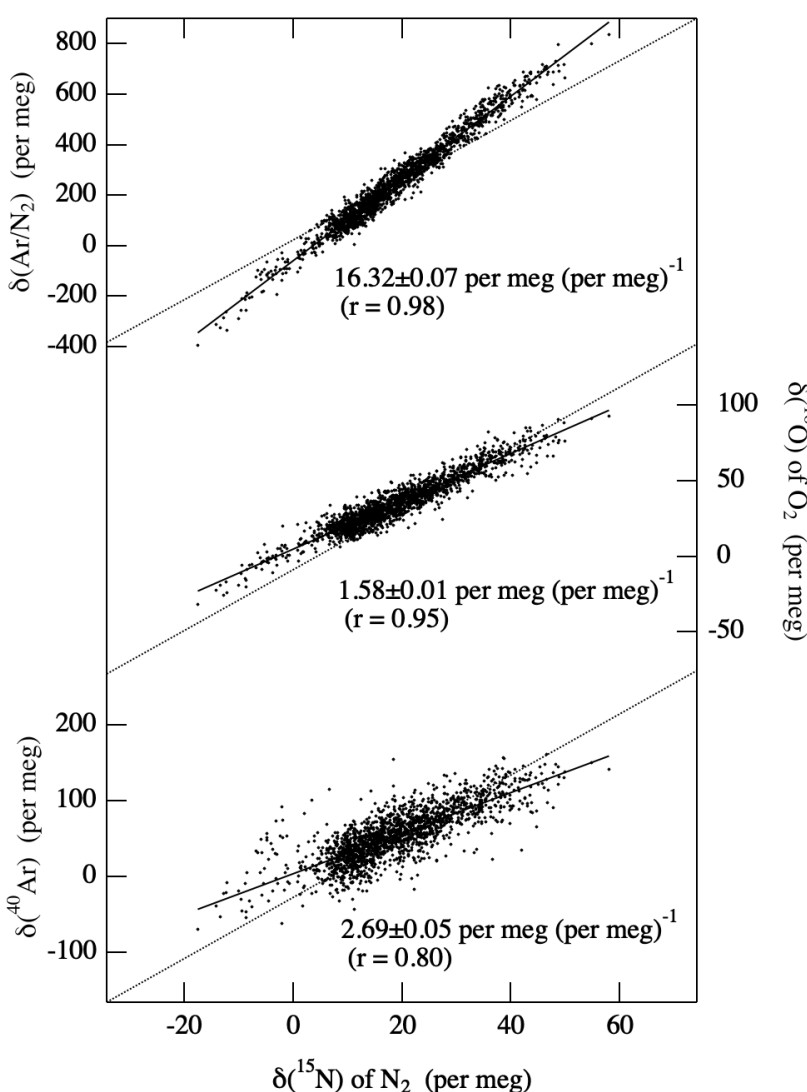

**Figure 2: Measured δ(Ar/N₂), δ(¹⁸O) of O₂ and δ(⁴⁰Ar) plotted against δ(¹⁵N) of N₂ for all the collected air samples (solid dots). Least-squares regression lines fitted to the data are shown as solid lines, while the relationships expected from mass-dependent fractionation of air molecules are shown by dotted lines.**

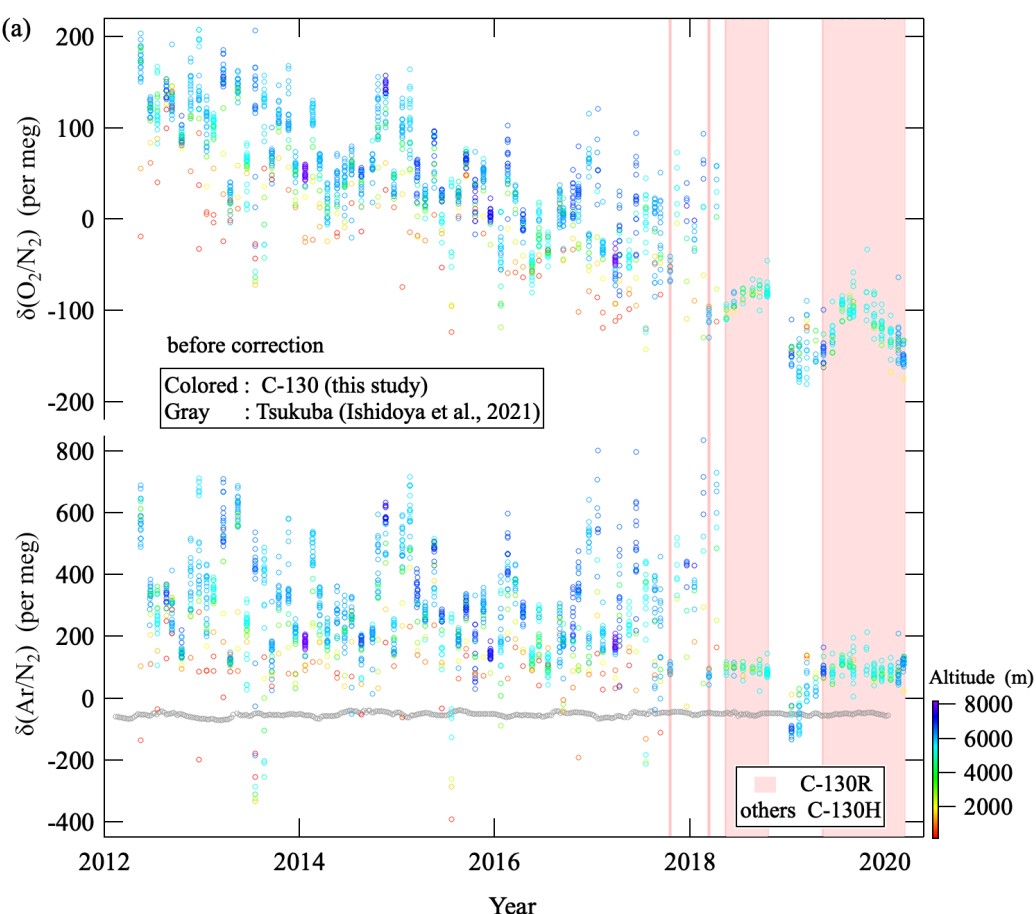

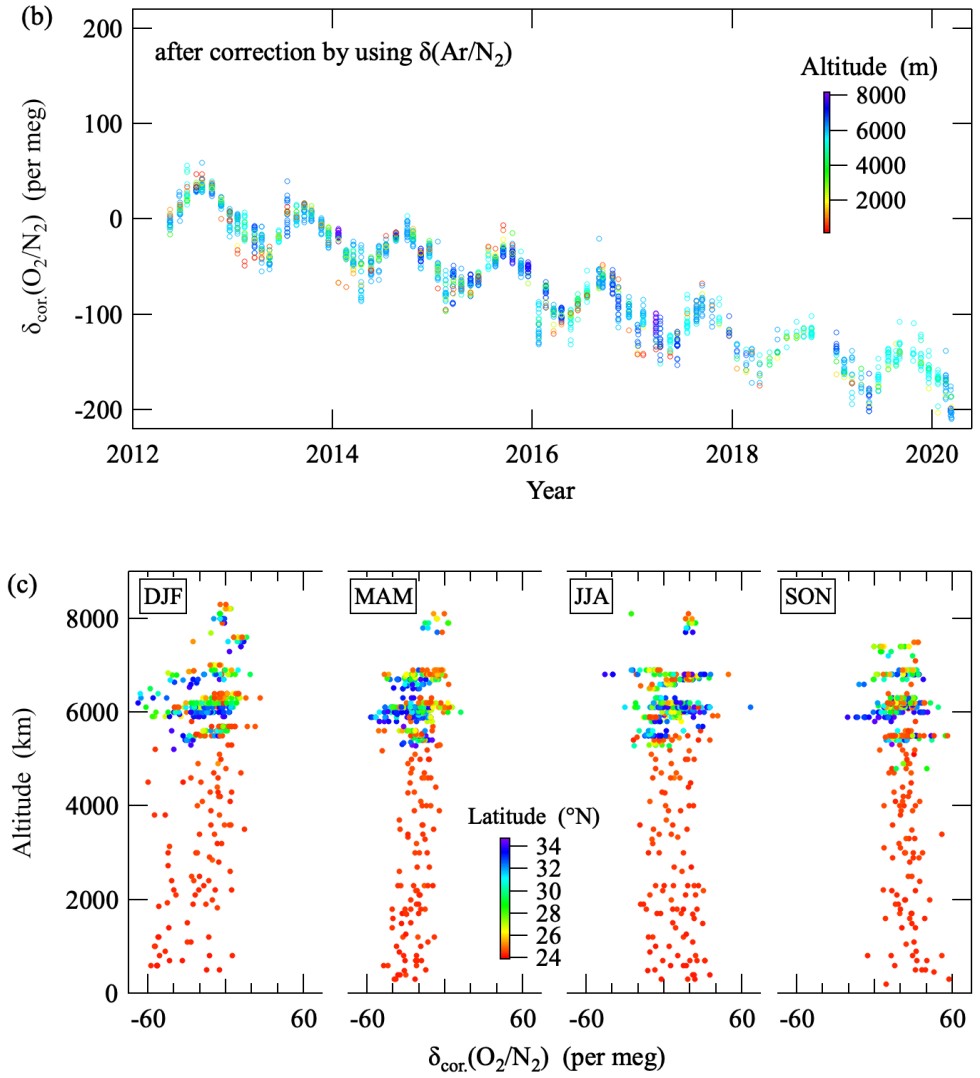

**Figure 3: (a) Measured values of $\delta(O_2/N_2)$ and $\delta(Ar/N_2)$ for all the air samples collected onboard the C-130 aircraft. $\delta(Ar/N_2)$ values observed at Tsukuba, Japan are also shown as gray circles (Ishidoya et al., 2021). Shaded areas denote the periods when C-130R aircraft was used, and C-130H aircraft was used in other periods. Color bar indicates the altitude where the air sampling were carried out. (b) $\delta_{cor.}(O_2/N_2)$ corrected for artificial fractionation by applying eq. (6) (see text). Color bar indicates the altitude where**
**the air sampling were carried out. (c) Vertical profiles of detrended $\delta_{cor.}(O_2/N_2)$ separated by season (DJF: December to February, MAM: March to May, JJA: June to August, and SON: September to November), obtained by subtracting a linear secular trend, fitted to the $\delta_{cor.}(O_2/N_2)$ in (b), from each $\delta_{cor.}(O_2/N_2)$ value. Color bar indicates the latitude where the air sampling were carried out.**

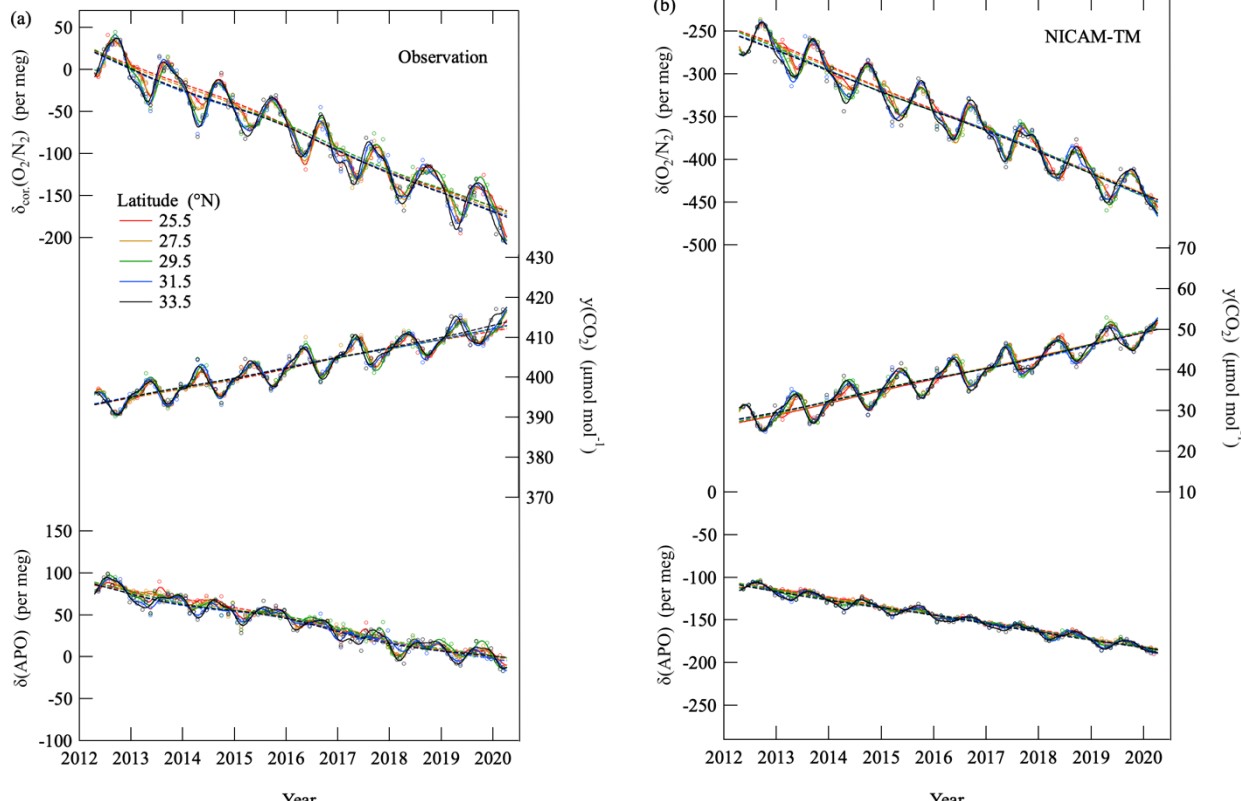

**Figure 4: (a)** $\delta_{cor.}(O_2/N_2)$**, CO$_2$ amount fraction and** $\delta$**(APO) observed at the altitude of (6.1 ± 0.5) (±1σ) km at various latitudes over the western North Pacific. Best-fit curves to the data (solid lines) and secular trends (dashed lines) are also shown (Nakazawa et al., 1997). (b) Same as in (a) but for calculated values obtained from the NICAM-TM control-run. The data from NICAM-TM are reported on arbitrary scales different from those of observations.**

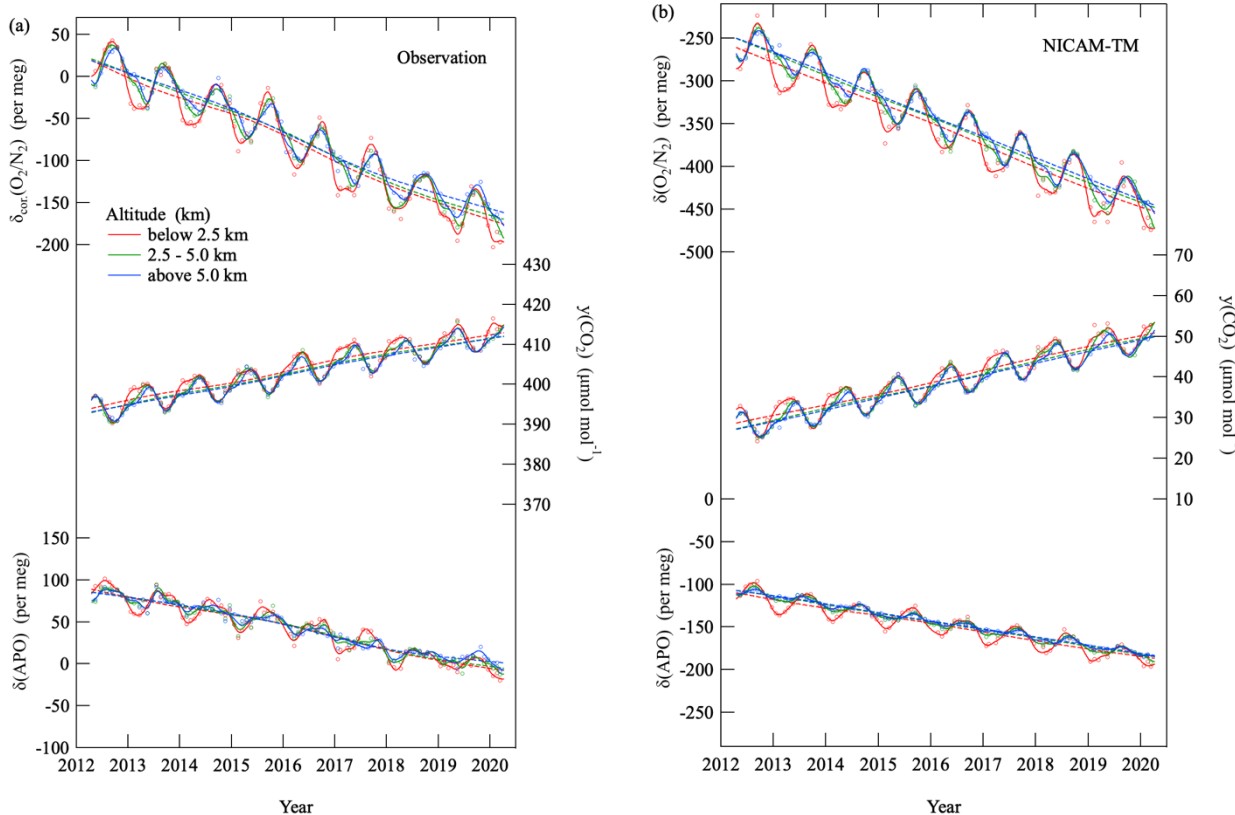

680

**Figure 5: (a)** $\delta_{cor.}(O_2/N_2)$**., CO$_2$ amount fraction and** $\delta$**(APO) observed in the troposphere over MNM at various altitudes. Best-fit curves to the data (solid lines) and secular trends (dashed lines) are also shown** (Nakazawa et al., 1997)**. (b) Same as in (a) but for calculated values using NICAM-TM. The data from NICAM-TM are reported on arbitrary scales different from those of observations.**

685

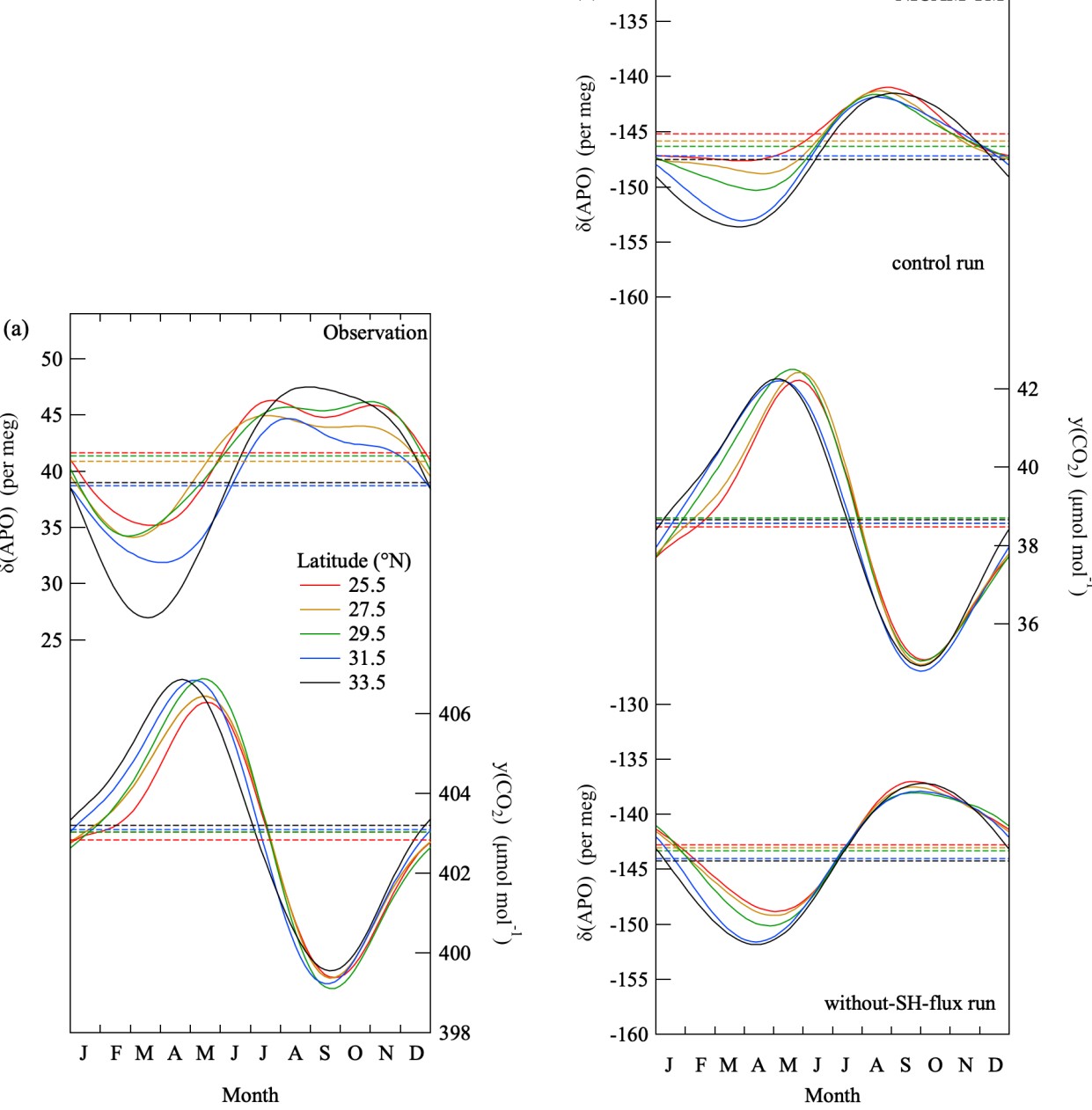

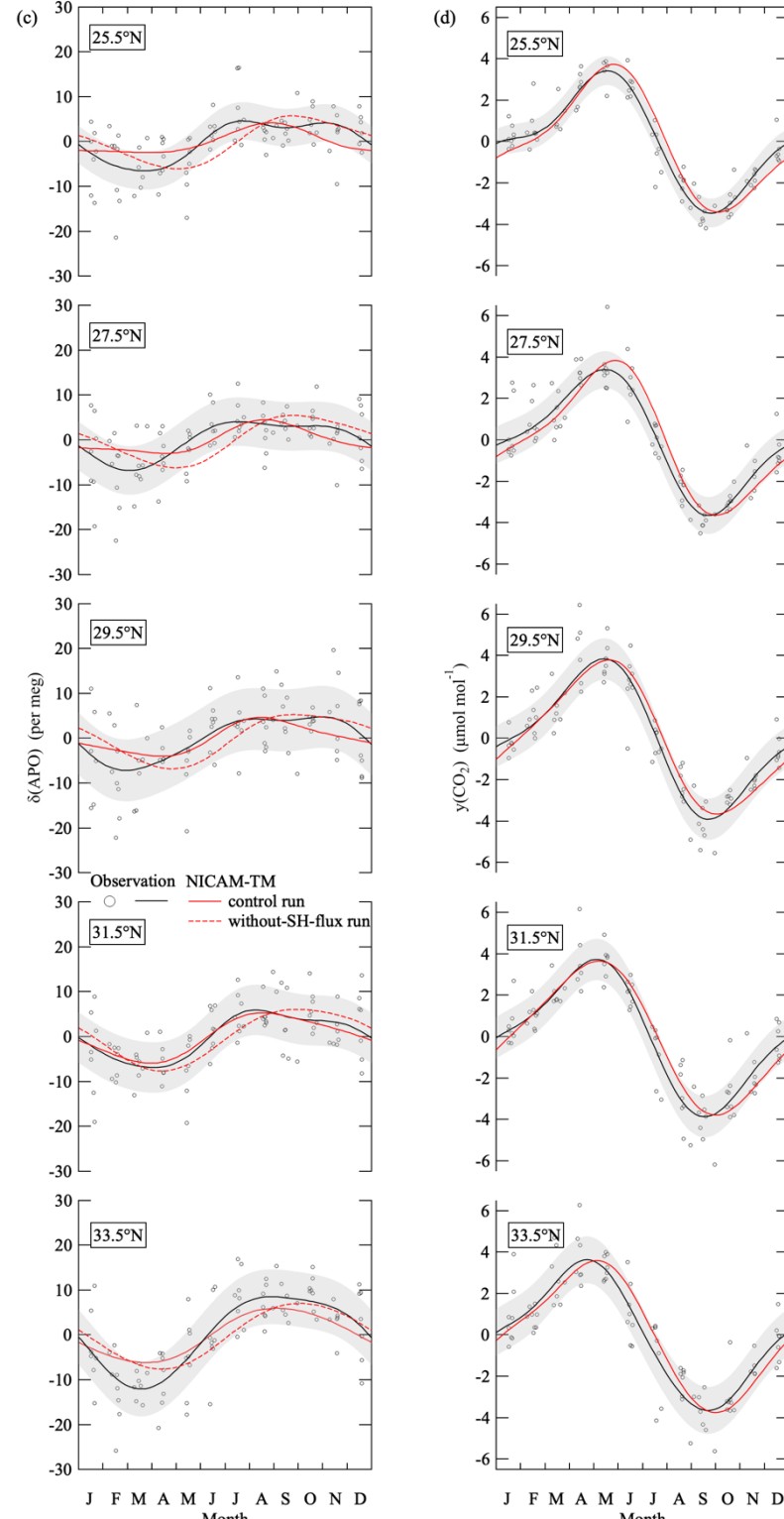

**Figure 6: (a) Average seasonal cycles of $\delta$(APO) and $CO_2$ amount fraction observed in the troposphere at various latitudes over the western North Pacific. Dashed lines denote the average values throughout the observation period. (b) Same as in (a) but for calculated values obtained from the NICAM-TM control run and the corresponding results from the without-SH-flux run (see text). (c) Same average seasonal cycles of the observed and simulated $\delta$(APO) in (a) and (b) along with the detrended observed values at each latitude. Error bands (shaded) indicate standard deviations of the observed $\delta$(APO) from the average seasonal cycles ($\pm1\sigma$). (d) Same as in (c) but for $CO_2$ amount fraction.**

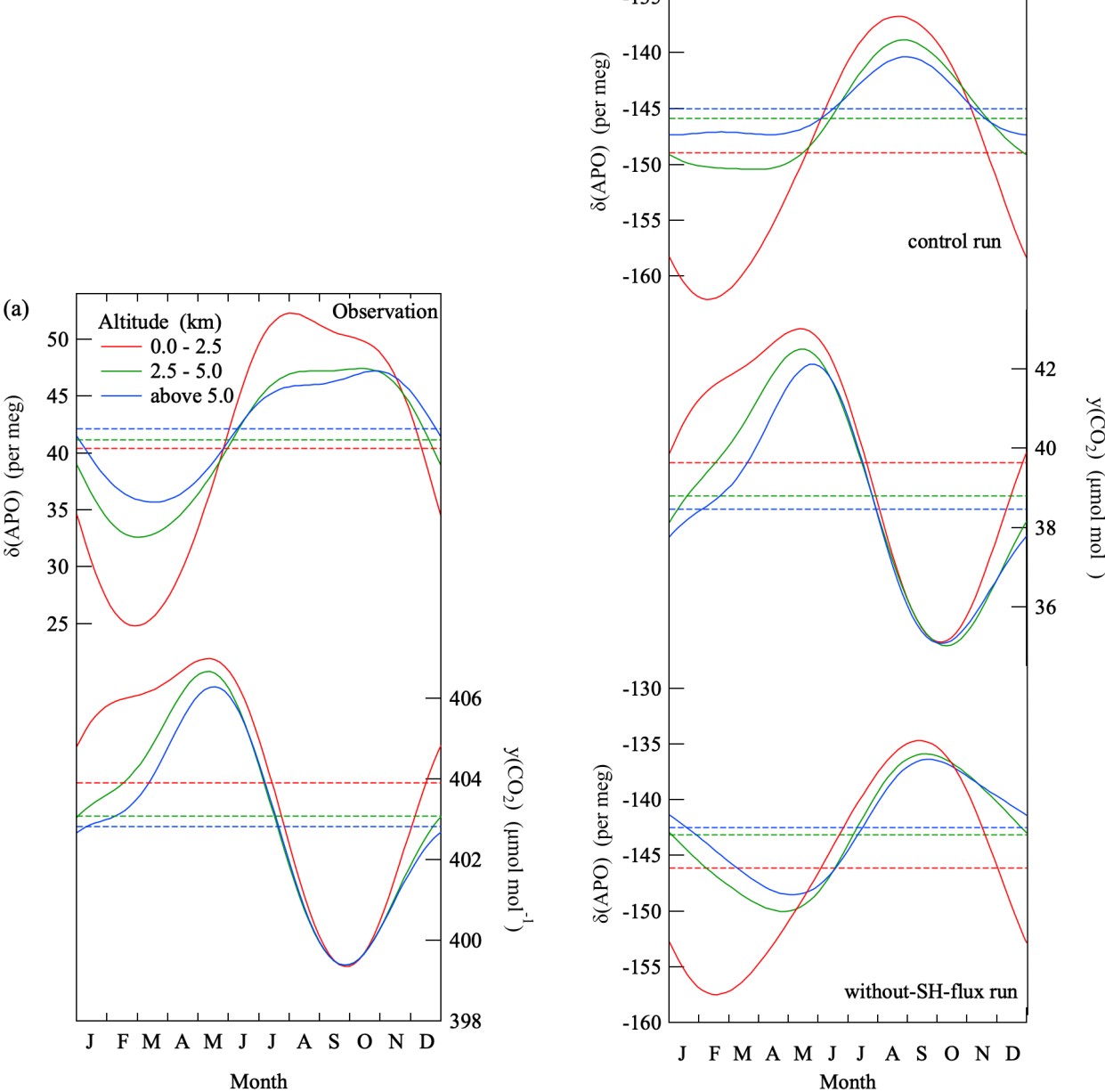

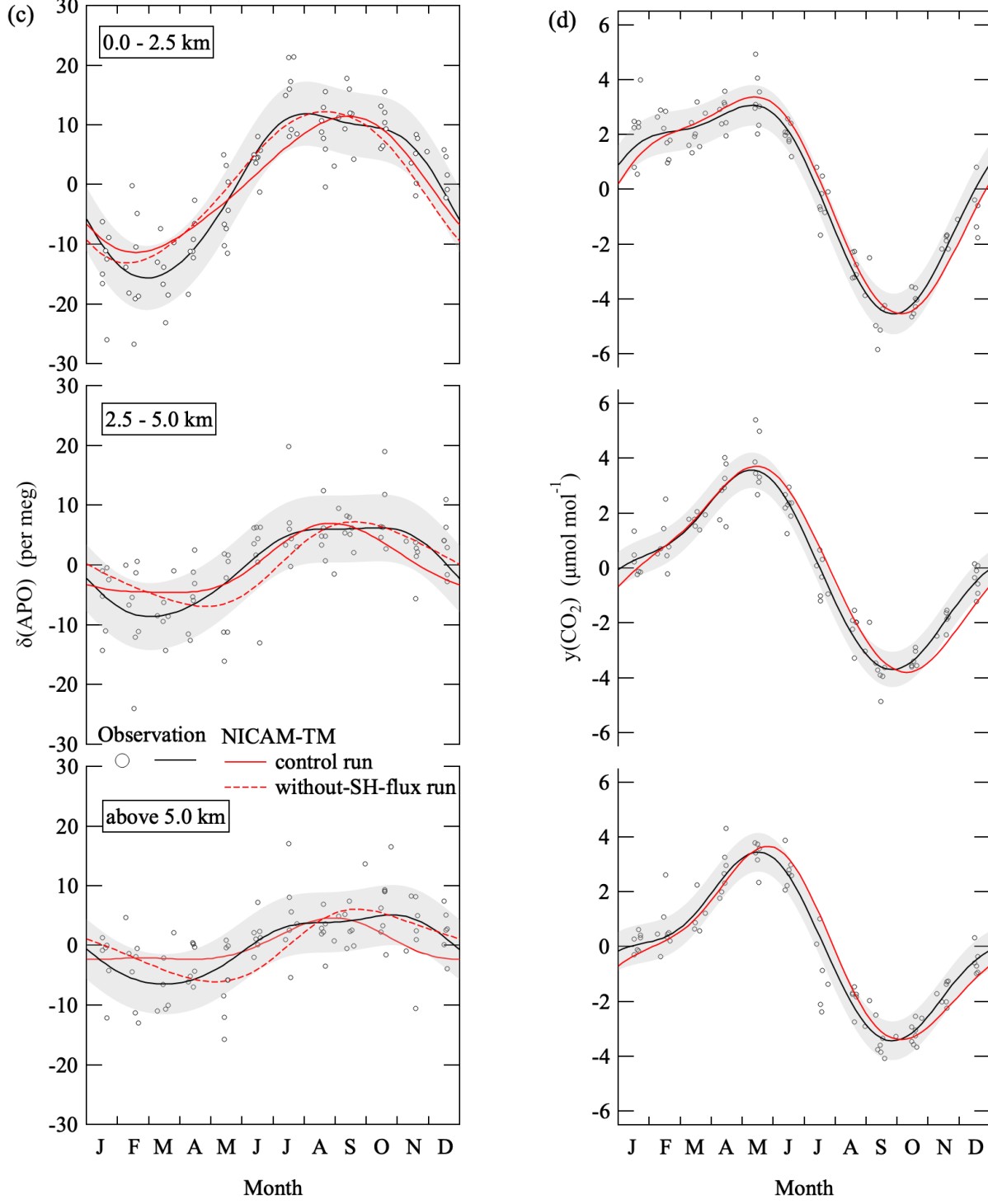

**Figure 7:** Same as in Fig. 6 but for those in the troposphere over MNM.

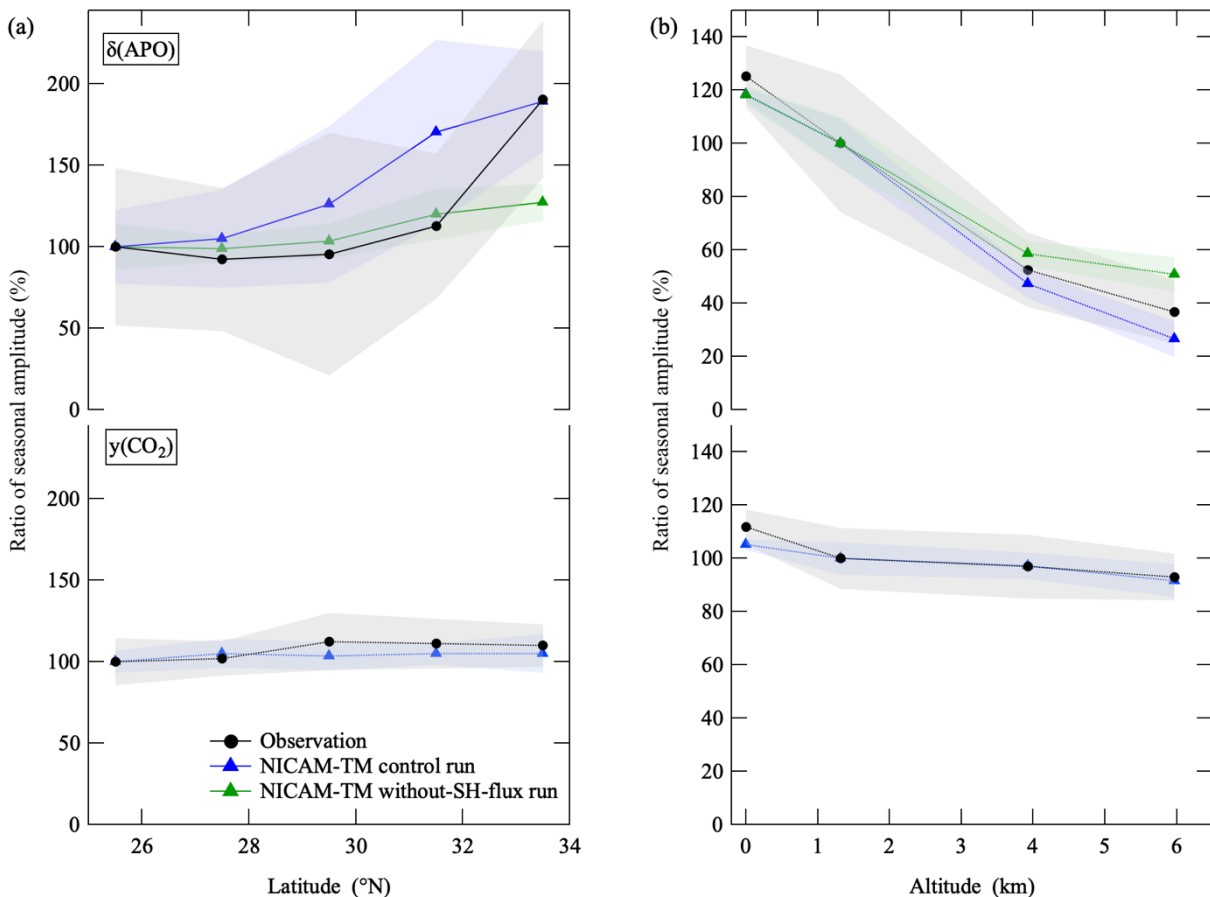

**Figure 8: (a)** Latitudinal distribution of average ratios of the observed seasonal $\delta$(APO) and $CO_2$ amount fraction amplitudes calculated relative to the values observed at 25.5° N in the troposphere over the western North Pacific throughout the observation period (black filled circles). Error bands (shaded) indicate year-to-year variations ($\pm 1\sigma$). The corresponding results calculated using NICAM-TM control run (blue triangles) and without-SH-flux run (green triangles) are also shown. Amplitude of seasonal APO cycle was calculated as a difference between an average value during July to September and that during February to April considering its broad peak, while that of $CO_2$ amount fraction was evaluated as a difference between a seasonal maximum and minimum values. **(b)** Same as in (a) but for vertical distribution over MNM relative to the corresponding values at 1.3 km, and amplitude of seasonal APO cycle was evaluated as a difference between an average value during July to October and that during February to April. The average fractions of surface seasonal cycles obtained from continuous observations of $\delta$(O$_2$/N$_2$) and $CO_2$

**amount fraction at MNM since December 2015, and the corresponding results calculated using NICAM-TM control run are also plotted.**

720

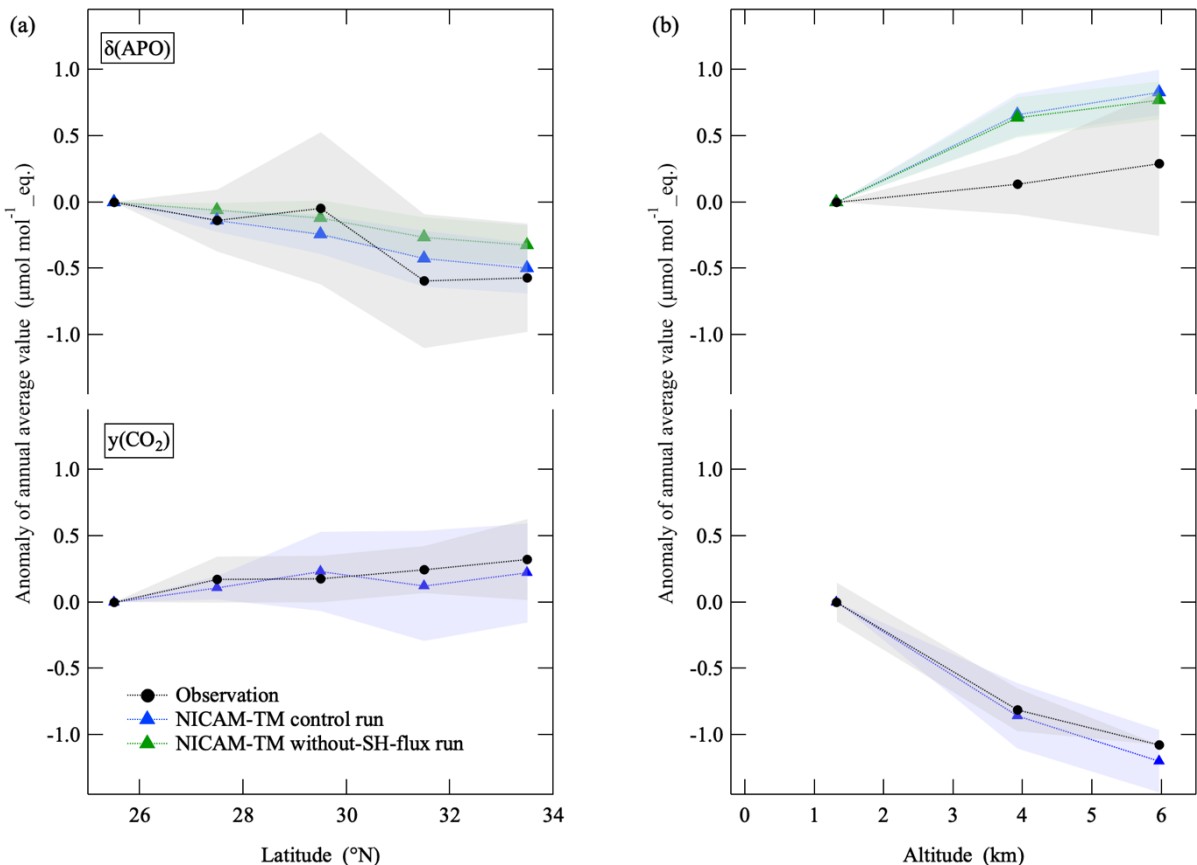

**Figure 9: (a)** Latitudinal distribution of average deviations of the annual mean values of δ(APO) and $CO_2$ amount fraction relative to those at 25.5° N in the troposphere over the western North Pacific throughout the observation period (black filled circles). Error bands (shaded) indicate year-to-year variations (±1σ). The corresponding results calculated using NICAM-TM for control run (blue triangles) and without-SH-flux run (green triangles) are also shown. **(b)** Same as in (a) but for vertical distribution over MNM relative to the corresponding values at 1.3 km.

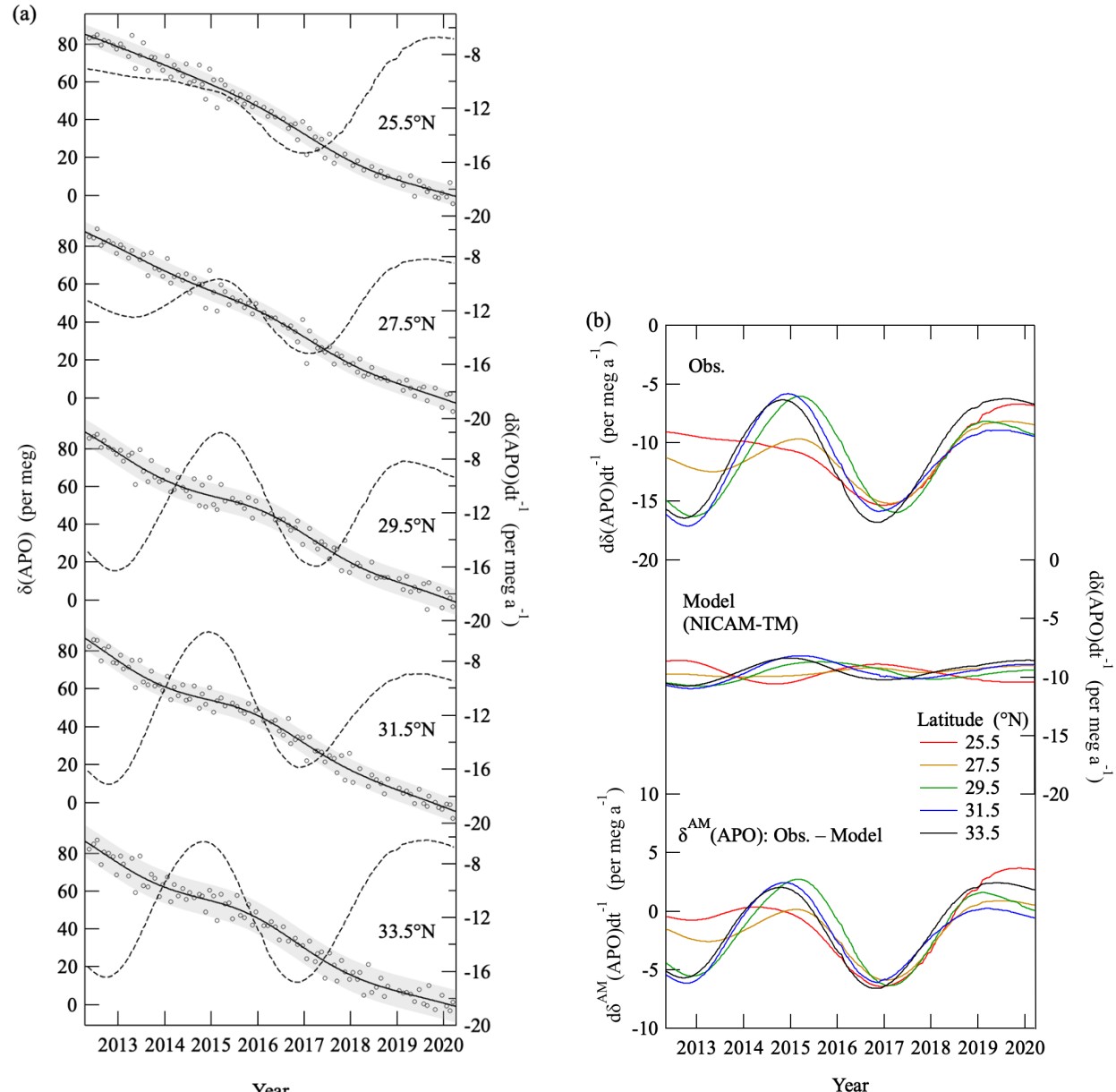

**Figure 10: (a)** Same secular trends in Fig. 4(a) and their annual change rates along with the deseasonalised values of $\delta$(APO) at each latitude over the western North Pacific. Error bands (shaded) indicate standard deviations of the deseasonalised $\delta$(APO) from the secular trends (±1σ). **(b)** Same annual change rates in (a) (top). The corresponding change rates for the APO obtained from the control run of NICAM-TM (middle) and those of $\delta^{AM}$(APO) obtained by subtracting the calculated $\delta$(APO) from the observed $\delta$(APO) (bottom) are also shown.

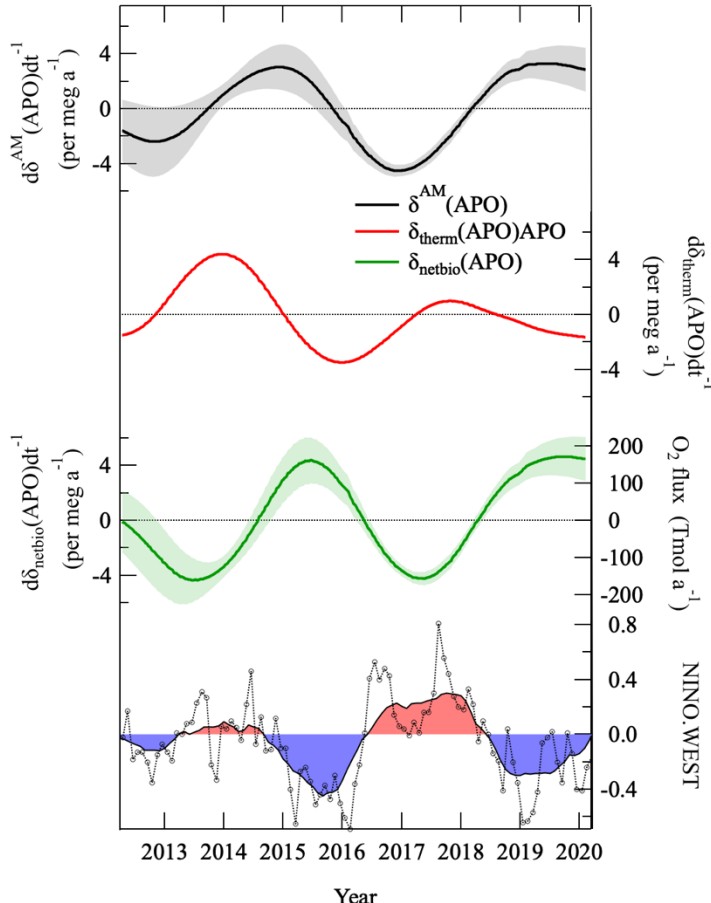

**Figure 11: Anomaly in the average annual change rate of $\delta^{AM}$(APO) shown at the bottom of Fig. 10 (thick black line). Anomaly in the change rate of APO driven only by the solubility change, expected from the observed surface $\delta$(Ar/N$_2$) at Tsukuba, Japan ($\delta_{therm}$(APO); red line, see text), and that driven by the net marine biospheric activities ($\delta_{netbio}$(APO); green line) obtained by subtracting the change rate of $\delta_{therm}$(APO) from that of $\delta^{AM}$(APO) are shown. Anomaly in the global air-sea O$_2$ flux corresponding to the change rate of $\delta_{netbio}$(APO) is also shown (see text). The time series of the NINO.WEST index (black open circles) and the annual average values of the index (thin black line) are shown at the bottom of the figure.**