# Peer review of "Spatiotemporal variations of the $\delta(O_2/N_2)$ , $CO_2$ and $\delta(APO)$ in the troposphere over the Western North Pacific"

_Atmospheric Chemistry and Physics, 2021_

## Author Comment (AC1)

**Responses to Referee 1**

General:

The manuscript presents new combined d(O2/N2) and CO2 values that allows the author to calculate Atmospheric Potential Oxygen (APO). Measurements of these parameters are obtained from flask samplings on board an aircraft between three different stations and an altitude transect at one of the stations. The measurements are analysed for their seasonality and secular trends and are compared to model results. The interpretation adds very valuable information for the understanding of the carbon-oxygen cycle links and helps to improve the budgeting of the global carbon cycle.

The manuscript is very nicely written with detailed information on how the method works and how it is used and applied to data. The figures and their legends are clear and concise.

It was easy to read the manuscript and I would like to congratulate the authors. I have only a few rather minor comments and suggestions. I suggest publishing it once these comments have been taken into consideration.

Thank you very much for your significant and useful comments on the paper "Spatiotemporal variations of the $\delta(O_2/N_2)$, $CO_2$ and $\delta(APO)$ in the troposphere over the Western North Pacific" by Ishidoya et al. We have revised the manuscript, considering your comments and suggestions. Details of our revision are as follows. The line numbers denote those of the revised manuscript.

Minor points:

Abstract: The corrections that are applied to the raw measurements are significant, how robust are these corrections. It is important that the reader gets already an impression of whether the corrections made are robust. I suggest rewording the sentence about the corrections by adding a corresponding statement about the robustness or adding an additional sentence about it.

Lines 192-196: The sentences have been added to discuss the robustness of the corrections.

Abstract: The altitude dependence of d(O2/N2), CO2 are not consistent percent-wise. This is obviously not the case for other locations. This should be discussed and compared to published studies about the altitude dependence in the corresponding section where the altitude dependence is mentioned. See also lines 2018-2019.

Lines 259-264: The sentences have been added to discuss the differences in the altitudinal dependence.

Line 111: Eq. 6 describes how you applied the corrections. Why is the correction based on Ar/N2 and not d15N, because you have excellent correlations with d15N and this parameter is stable in the atmosphere over long time periods?

Lines 139-142: We agree that the correction based on $\delta^{15}N$ is also suitable. Therefore, we have added

following sentence. "It is noted that the correction by using $\delta(^{15}N)$, which is stable in the atmosphere over long time period, is also suitable to obtain $\delta_{cor.}(O_2/N_2)$. Since the uncertainty of the corrected $\delta_{cor.}(O_2/N_2)$ by using $\delta(^{15}N)$ is ±7 per meg (±1.5 x 4.57), which is larger than that by using $\delta(Ar/N_2)$ (±2 per meg), we determined to use $\delta(Ar/N_2)$ rather than $\delta(^{15}N)$ in the present study".

Line 113: The value for aO2 = (4.57±0.02) is not directly reported in Ishidoya, you may refer here to how you calculated.
Lines 126-127: The sentence has been added to refer to how we obtained the value.

Line 116: The overall uncertainty of dcor(O2/N2) was evaluated to be less than 6 per meg, and the effect of the seasonal d(Ar/N2) cycle on of dcor(O2/N2) was not therefore excluded in this study. This sentence is unclear to me.
Lines 130-135: The sentences have been revised to make the meaning clearer.

Line 285: Fig. 11 instead of Fig. 12.
Line 331: "Fig. 12" has been changed to "Fig. 11".

Fig. 1:       One could indicate in this graph that at MNM altitude profiles are taken.
Fig. 1, caption: The sentence "Latitudinal and vertical distributions are taken during the level flight and descent portion at MNM, respectively" has been added.

Fig. 10:       It is not clear how the rate change values on the top panel of Fig. 10 are obtained. The values should be positive and negative. What about uncertainties. The spline functions in Figure 4 have uncertainties associated, could you add shading on the derivatives (e.g. Fig. 10) to illustrate these uncertainties for readability reasons only for one curve.
Fig. 10: Fig. 10(a) has been added to clarify the relationship between the secular trends of APO and its rate change values. Uncertainties of the secular trends can also be seen from the error bands in the figure. Since APO decreases secularly, the rate change value is generally negative.

Other changes
Lines 146-147: The sentence has been modified and Tohjima et al. (2005) has been added to reference since we have noticed that we used $X_{O_2}$ of 0.2094 in their study to calculate the observed $\delta(APO)$.

Lines 201-203, Figs. 4 and 5: The sentence has been added to note the data selection in the digital filtering technique, and the observational data deviated from the best-fitted curves more than ±3σ have been excluded from Figs. 4 and 5.

Figure 8 caption: The sentence to show the method to calculate the amplitude of seasonal APO and $CO_2$ cycles have been added.

---

## Author Comment (AC2)

**Responses to Referee 2**

General comments:

Ishidoya and colleagues present ~8 years of airborne observations of O2/N2, CO2, and Ar/N2 from cargo aircraft flights over the North Pacific. The data are significantly impacted by sampling artifacts. The authors correct this data convincingly, and, from the corrected data, calculate the apparent global ocean and terrestrial CO2 sinks. They also show that the seasonal cycle in APO is influenced by interhemispheric transport. An analysis of the interannual variability in the observations shows a signal which can be attributed to ENSO.

I think it could be made clearer in the text that the transported fluxes are derived from climatologies, which have known deficiencies. The Garcia and Keeling fluxes, for instance used the Wanninkhof 92 gas transfer velocity which is now known to be biased. The authors might consider using potential temperature or pressure as a vertical coordinate for binning instead of altitude.

I think the study is well conducted and worthy of inclusion in ACP. I have raised some minor points to be addressed below.

Thank you very much for your significant and useful comments on the paper "Spatiotemporal variations of the $\delta(O_2/N_2)$, $CO_2$ and $\delta(APO)$ in the troposphere over the Western North Pacific" by Ishidoya et al. We have revised the manuscript, considering your comments and suggestions. Details of our revision are as follows. The line numbers denote those of the revised manuscript.

ABSTRACT

L11 and throughout: "amount fraction" is not a term I'm familiar with, I suggest "abundances" for referring to both O2/N2 and CO2, and "mole fraction" for referring to CO2.

We recognize the phrases you suggested are more familiar with our research field, however, I have used the phrase following the Editor's comment.

L11: "Observations were corrected for significant..."

Line 11: We have modified the sentence, as suggested.

L18 and throughout: usually "northern hemisphere" and "southern hemisphere" are capitalized, but I would refer to the specific style guidelines of the journal.

We leave the words as they are. If the Editor requests the revision before publication, then I will revise the words to be capitalized.

L22: "indicated a clear evidence of influence" -- suggest change to "indicated a clear influence"

Lines 21-22: The words have been changed to "indicated a clear influence", as suggested.

L24: What is a "C equivalent"? Do you mean simply petagrams of carbon? If so, Pg C a-1 is a widely used unit.

We recognize the unit you suggested are more familiar with our research field, however, I have used the unit following the Editor's comment.

INTRODUCTION

L27: "ratio" should be "ratios"

Line 27: The words "ratio has" have been changed to "ratios have", as suggested.

L27: "marine biospheric activities" is a little unclear to me. I might suggest simply "...early 1990s, for the primary application of constraining the marine and terrestrial exchange of CO2."

Lines 27-28: The sentence has been modified, as suggested.

L31: suggest "terrestrial biosphere exchanges"

Lines 31-32: The words have been changed to "terrestrial biosphere exchanges", as suggested.

L36: suggest "carbon dissociation effect" read something like "the carbonate buffer system"

Line 36: The words have been changed to "the carbonate buffer system", as suggested.

L39: I think there is a missing sentence here explaining that airborne observations are useful because they quickly map a large spatial area. Suggest cutting "from this point of view" and moving the sentence to the first sentence of the paragraph beginning with "Aircraft observations"

Lines 39-40: The sentence has been moved to the first sentence of the paragraph, as suggested.

L40-45: Steinbach (2010) might be worth citing here, since it pre-dates the Ishidoya and van der Laan references. Bent (2014) might be worth citing here as well, since he also reported Ar/N2.

Lines 41: Steinbach (2010) and Bent (2014) have been cited, as suggested.

METHOD

This section could use subsections for easier reading.

L76: Suggest changing "Method" to "Methods" in the section heading.

Line 36: The word "Method" has been changed to "Methods", as suggested.

L78: Suggest rewording to "Minamitorishima, Japan, a small coral atoll (MNM; 24.28N, 153.98E)".

Line 78: We have modified the sentence, as suggested.

L78: "The cruising altitude is about 6km"

Line 78: The words "flight altitude" have been changed to "cruising altitude".

L79: "titanium" (not capitalized)

Line 79: "Titanium" has been changed to "titanium".

L101: Could the authors specify the scales these species are measured on, at least for O2/N2 and CO2?

Lines 93-95 and Lines 112-113: The sentences have been added to specify the scales, following your suggestion.

L102-103: The authors cite Ishidoya et al 2014, which cites Niwa et al 2014, which cites Tsuboi et al 2013. I will admit I only scanned the papers but it seems Tsuboi is the only refernce that describes the intake and flask sampling apparatus. So I would point directly to this paper on L81 to save the reader time. This paper does not seem to have a diagram of the flask sampler, so unless I missed it in one of these papers I think it would be nice to include either in the text or in a supplement. I think this is important because the fractionation of the samples is quite considerable. I am still not clear on what kind of inlet is actually outside the airfract. From Stephens et al 2021 it was clear that the design and orientation of the inlet is critical for avoiding fractionation. Where is the air conditioning inlet located on the plane, and what does it look like? The thermal fractionation the authors identify is so massive it is probably obscuring other sources of artifacts, like at the inlet or somewhere in the air conditioning system. Since sample air passes through this, could the authors include it in the plumbing diagram? I am also surprised the authors don't have serious problems with surface effects, given that the teflon tubing is used, flasks are only partially dried, they are pressurized fairly high to 0.4 MPa, and then analyzed (I think) without a push gas. I am sure the authors have worked all this out, and probably have already published details on it, but without details here or specific citations it's hard to understand the sampling conditions.

Lines 80-91: We have changed the sentence as "Details of the air sampling method has been described elsewhere (Tsuboi et al., 2013)" to point directly to the paper (lines 90-91). Unfortunately, details of the air sampling line from the inlet to flask sampler have not been informed to researchers from Japan Ministry of Defense. Therefore, that was the only thing we can do was to add some information from Tsuboi et al. (2013) and an issue associated with the using o Teflon tubing (Lines 80-90). As to drying, water vapor is removed by using not only $Mg(ClO_4)_2$ during air collection but also a cold trap at -50°C during analyses. The analyses have been carried out without a push gas, as you expected. Consequently,

the relationships between the measured $\delta(Ar/N_2)$, $\delta^{18}O$ and $\delta^{40}Ar$ with $\delta^{15}N$ show total fractionation including the processes you pointed out.

L113: Perhaps cite here that the ratio of the scaling factors 4.57/16.2 is close to the Keeling et al 2004 diffusion factor for (Ar/N2)/(O2/N2) and results in the same tracer d(O2/N2)*
Lines 127-128: The sentences have been added considering your suggestion.

L115: Since all of the samples are from the same region, why not use the monthly mean at Tsukuba? I expect there will be some lag to consider, but otherwise it seems like this is introducing an unnecessary approximation.
Lines 130-132: Your suggestion is reasonable, but we leave the correction processes as they are. In this study, we simply evaluate the uncertainty associated with the seasonal $\delta(Ar/N_2)$ cycle since the spatial variations in the surface $\delta(Ar/N_2)$ have not been clarified yet sufficiently, and as you pointed out, there is some lag due to atmospheric transport to be considered.

L116: How was the uncertainty evaluated, and what terms are contributing to the total uncertainty? Are you accounting for natural variations in Ar/N2? Is this what is meant in L117? How much does the annual mean of Ar/N2 vary? Have the authors considered forcing to a constant value of Ar/N2? This might be preferable since the paper deals with interannual trends.
Lines 130-138: The sentences have been added to discuss the matters you pointed out.

L119: This seems like the beginning of the "Results" section to me, since data is presented.
Lines 185-196: We have moved the sentences to the beginning of the "Results and discussion", as suggested.

L119-125 and Fig3: Could the authors include a panel showing the vertical profiles of detrended (O2/N2)cor, perhaps separated by season? It is hard to evaluate the quality of the data when only shown as a time series. Also, I don't see what the bottom plot of panel b in Fig 3 is adding, since the data are also shown in the bottom plot of panel a. The authors could replace this with profiles or I change the y axis of Ar/N2, since there is not much that can be seen at that scale. Is it correct that the red line is the annual mean Ar/N2 for the Tsukuba time series?
Lines 192-196, and Fig. 3(c): We have added Fig. 3(c) showing the vertical profiles of detrended $\delta_{cor.}(O_2/N_2)$ separated by season, and modified Fig. 3(b), as suggested.

L124: It would be good to indicate which samples were taken with which type of aircraft with a vertical line or some other indicator. Is it correct that only two aircraft were used? Could this be given in the

Methods? It appears that the change caused the thermal fractionation effect to be reduced by more than 400 per meg.

Line 190, and Fig. 3(a): Fig. 3(a) have been modified to show the periods when the C-130R were used by pale red shade. Some aircrafts are used as C-130H and C-130R, respectively (i.e. not limited two just two aircraft).

L150: is the "seasonal anomaly" of O2 the Garcia and Keeling 2001 climatology?

Line 167: The words "the seasonal anomaly" have been changed to "the seasonal anomaly of the air-seas $O_2$ and $N_2$ flux", to make the meaning clearer. The flux is from Garcia and Keeling (2001), as you expected.

L153: Where does the 1.35 value for the global OR for fossil fuel combustion come from? It is lower than the values given in Keeling and Manning 2014 and I think lower than what the CDIAC data would suggest.

Lines 170-172: We have corrected the global OR for fossil fuel combustion to 1.37 calculated based on fossil emission by category summarized in GCP (Friedlingstein et al., 2020).

L156: I don't understand this sentence: "driven by an annual mean air-sea O2 and N2 fluxes...that was considered by Tohjima...was ignored". Also, in the Tohjima et al 2012 reference there is an unnecessary hyphen in "annual". In the Tohjima paper this seems refer to the Gruber et al ocean inversion O2 fluxes? Was there a separate run of the Gruber fluxes? Or just two products: 1) Garcia and Keeling + Takahashi + CDIAC and then 2) simulated - observed APO?

Lines 176-178: The sentence has been changed to "In this calculation, $\delta$(APO) driven by an annual mean air-sea $O_2$ and $N_2$ fluxes (hereafter referred to as the "$\delta^{AM}$(APO)"), which were estimated by Gruber et al. (2001), was ignored", considering your comment. The typo in the reference has been corrected. Thank you for pointing out.

RESULTS

I suggest to cut some of the L161-177 text, the decrease in O2/N2, the rise in CO2, and their seasonal cycles are well known. I would start the paragraph at "The average rates of change..." with figure citation.

We understand your suggestion, but we leave the sentences as they are, for the convenience of the readers who are not familiar with the $O_2/N_2$ study.

L171/Fig4: there are multiple fit lines to the data but this is not explained in the caption. Maybe a legend? I think it

I think we have explained the fit curves as "Best-fit curves to the data (solid lines) and secular trends (dashed lines) are also shown" in the caption of Fig. 4(a).

L186/Fig 6: I think it would be better to plot each latitude bin as a separate plot with observations and model together, it is a little difficult to compare as is. It would also be nice to see the seasonal cycle in the detrended observations along with the fits.

Lines 223-225, Lines 229-230, and Figs. 6 and 7: We have added not only Fig. 6(c) and (d) but also Fig. 7(c) and (d) to plot each latitude bin as a separate plot with observations and model together, as suggested. The seasonal cycles in the detrended observations along with the fits have also been added in the figures. Related sentences have been also added in the text.

L194: Suggest changing "anti-phase nature in the seasonal APO cycles" to the "opposing phase of the seasonal APO cycles"

Line 235: The words "anti-phase nature" have been changed to "opposing phase", as suggested.

L197: I don't understand "superimposing the anti-phase seasonal cycles through the inter-hemispheric mixing of air". From this I would think you are running only the SH flux, but the "w/o SH flux run" would imply it was northern hemisphere fluxes only. Earlier in L192 it says "northern hemisphere flux only".

The "w/o SH flux run" means northern hemisphere flux only in the seasonal air-sea $O_2$ and $N_2$ fluxes, which yields significant seasonal APO cycle in the northern hemisphere only. On the other hand, in the "control run", the anti-phase seasonal cycles in the northern and southern hemisphere are superimposed through the inter-hemispheric mixing of air in the lower latitude. We examined an effect of the inter-hemispheric mixing of air from the comparison between the "control run" and "w/o SH flux run".

L198/199: should read "the seasonal cycle in CO2 mole fraction" or just "seasonal cycle in CO2".

Line 240: The words "seasonal $CO_2$ amount fraction cycle" have been changed to "the seasonal cycle in $CO_2$ amount fraction". We leave the phrase "amount fraction" as it is following the Editor's comment.

L205: From figure 6 it looks to me that most of the seasonal cycle is due to NH fluxes, as one would expect...it does not look that much smaller to me. Can you give the amplitudes of the two runs?

From Fig. 6(b), the peak-to-peak amplitudes of the seasonal APO cycle are 8.7, 10.0, 11.6, 13.8 and 15.8 per meg at 25.5, 27.5, 29.5, 31.5 and 33.5 °N, respectively, for the "control run", while those are 12.1, 11.9, 12.5, 14.5 and 15.4 per meg for the "w/o SH flux run".

L213-215: I don't fully follow this--it's the gradients in the fluxes, with contributions from atmospheric transport, that causes a gradient in the amplitude in the atmosphere. I think it would be clearer to say simply that the SH makes a significant contribution to the amplitude and phase of the lower latitude observations. I would also caution against over interpreting the model results--the transport model could be over or underestimating the interhemispheric transport.

Lines 254-255: The sentence has been rewritten as "These results indicate that the SH makes a significant contribution to the amplitude and phase of the seasonal δ(APO) cycle at lower latitude", considering your comment.

L224: Suggest changing "highly" to "more"

Line 269: The word "highly" has been changed to "more".

Fig8/9: Typically one plots altitude on the y axis, but I leave it up to the authors.

Thank you for suggestion, but we leave the figures as they are in the present study.

Fig10: This looks like a fit to the data, can you include the points as well? The data look a little odd, I would expect observations of the APO growth rate to look noisier.

Lines 289-290 and Fig. 10 (a): The figure has been added to show the deseasonalised values of δ(APO) at each latitude with the secular trends. Uncertainties of the secular trends can also be found from the error bands in the figure, which will also be useful to imagine the uncertainties in the change rates.

L265: This is a rough approximation of the thermal component of APO, which is a combination of air-sea fluxes of O2, N2, and CO2 caused by solubility changes. The "netbio" APO will also have a contribution from fossil fuel burning/CO2.

Lines 312-313: The sentence has been added to note the thermal component of APO estimated in this study is a rough approximation, considering your comment.

L273-287: I am not fully convinced that this exercise accomplishes more than very roughly constraining the global APO and O2 flux. The number of simplifying assumptions is extensive, and using aircraft observations from a comparatively small region, sparse data coverage, and enormous sampling artifacts is not an ideal approach. To me what this shows is that the corrected data have a coherent signal the authors can explain, and helps to prove that the data are of good quality. But I would caution against over interpreting and overselling the data.

Lines 334-337: The sentences have been added to include your suggestion and caution in the text.

L301: I thought 1.35 was used?

As mentioned above, we have corrected the global OR for fossil fuel combustion to 1.37 calculated based on fossil emission by category summarized in GCP (Friedlingstein et al., 2020).

L312: Where does the 4-5 year period come from? Just visual inspection? Missing here is a citation for the Nevison et al 2008 study, which (also) showed that the errors from assuming a constant zeff over short time scales (e.g. 5 years) are quite large.

We have added Nevison et al. (2008) to the reference (line 362). Tohjima et al. (2019) also examined the period based on their observational data (Fig. 7 in their study).

CONCLUSIONS

I would change this section heading to "Summary"

Line 375: "Conclusions" has been changed to "Summary".

L327: Better something like "Regular air samples were taken on cargo aircraft flights from..."

Line 376: The words "Cargo aircraft C-130 flies once per month from Atsugi Base to MNM" have been changed to "Regular air samples were taken on cargo aircraft C-130 flights from Atsugi Base to MNM".

L341 and throughout: "Superposition" is a slightly strange choice of words, I might suggest something simple like "combination".

"superposition" has been changed to "combination", as suggested.

Other changes

Lines 146-147: The sentence has been modified and Tohjima et al. (2005) has been added to reference since we have noticed that we used $X_{O2}$ of 0.2094 in their study to calculate the observed $\delta$(APO).

Lines 201-203, Figs. 4 and 5: The sentence has been added to note the data selection in the digital filtering technique, and the observational data deviated from the best-fitted curves more than $\pm3\sigma$ have been excluded from Figs. 4 and 5.

Figure 8 caption: The sentence to show the method to calculate the amplitude of seasonal APO and $CO_2$ cycles have been added.

---

## Author Comment (AC3)

**Responses to Referee 3**

general comments

This manuscript presents 8 years of d(O2/N2) and CO2 observations and calculated APO values from aircraft flights over the North Pacific. The data were corrected for significant fractionation effects on O2 and N2. The data were then well-analysed to find latitudinal and altitudinal, seasonal and secular trends in APO and the authors have demonstrated the influence of inter-hemispheric mixing on the seasonal APO cycle through comparison to model data. This manuscript is well-designed and well-written, and the discussion and interpretation of results contributes to the understanding of global atmospheric carbon and oxygen processes. I can recommend this manuscript for publication in ACP, with some minor comments below.

Thank you very much for your significant and useful comments on the paper "Spatiotemporal variations of the $\delta(O_2/N_2)$, $CO_2$ and $\delta(APO)$ in the troposphere over the Western North Pacific" by Ishidoya et al. We have revised the manuscript, considering your comments and suggestions. Details of our revision are as follows. The line numbers denote those of the revised manuscript.

specific comments

Line 24: Units are usually written as Pg C a-1, if this is what is meant by C equivalents

We recognize the unit you suggested are more familiar with our research field, however, I have used the unit considering the Editor's comment.

Line 24: I suggest that here, and elsewhere, a space should be added between the number ± and the uncertainty value for ease of reading. E.g. "1.9±0.9" changed to "1.9 ± 0.9". I would also suggest removing the brackets around all of your quoted values, especially as the units are stated outside of the brackets.

A space has been added between the number ± and the uncertainty value throughout the paper, as suggested. As to the brackets, I understand your suggestion, but we followed the Editor's comment.

Line 31: Here, and throughout, you have referred to CO2 amount fraction – this is usually referred to as CO2 mole fraction

We recognize the phrase you suggested are more familiar with our research field, but we followed the Editor's comment.

Line 81: Although the sampling methods are described in full in the stated references, I think a brief summary of a few lines would be helpful to the reader here and then direction to the references for a full detailed description

Lines 80-91: The sentences have been added to show a brief summary of the sampling methods and related information, as suggested.

Line 83: What scale is the CO2 measured on?

Lines 93-95: The sentence has been modified to show the scale of $CO_2$, as suggested.

Lines 88-90: at some point here, or in the figure 1 caption you could include how many air samples were collected in total. You have said that 17-20 are collected per flight, but not how many flights total.

Figure 1 caption: We have added the information about the number of air samples collected and analyzed.

Lines 93-97: While these equations are correct, I would suggest writing them in full i.e. (sample-standard)/standard. The form shown here is a mathematical simplification and results in some loss in understanding of the principal behind the equation

Lines 104-108: The equations have been rewritten, as suggested.

Line 99: Which scale have each species been calibrated to?

Lines 112-113: We have added the information about the scale.

Line 116: how was this overall uncertainty calculated? Is this from the measurement uncertainty and the stated uncertainties in the coefficients from equation 6?

Lines 130-135: The sentence has been added to show the overall uncertainty, as suggested.

Line 117: I found the phrase "was not therefore excluded in this study" difficult to comprehend. I would suggest rewording to "was therefore not excluded in this study", or "was therefore included in this study"

Line 131: The phrase has been changed to "was therefore not excluded in this study", as suggested.

Figure 3: The legend on 3(a) shows this studies data as an open red circle, whereas in the figure I am assuming that they are coloured by altitude (as they are in 3b), I would suggest adding the altitude colour bar to 3(a) also. The bottom panel of 3(b) is not referred to in text and is showing the same data as the bottom panel on 3(a) so could be removed if the red reference point line were added to the bottom panel of 3(a). Is the red line reference point of d(Ar/N2) the annual mean value from Tsukba in 2013? If so this information could be added to the figure caption for further clarification, if not, what is it?

Figure 3: We have made substantial revision of Fig. 3, considering your and the other reviewers' comments.

Line 123: I don't think "but" is the correct word here, as that implies that the reduction in fractionation since 2018 is linked to the larger fractionations at higher altitudes before 2018 – unless this is the case, and if so this should be reworded to make this clearer
Lines 187-188: The sentence has been modified to make the meaning clearer, as suggested.

Line 124-125: The word "however" implies that the lack of systematic data gaps across 2018 mean that the change in aircraft may not be the cause of the reduction in fractionation, I don't understand this. If this is the case, could you suggest another cause of this reduction in fractionation - is the change in aircraft the only change that occurred in 2018? The reduction in fractionation is substantial so further discussion of this would be useful.
Lines 190-192: The sentence has been rewritten as "No systematic data gaps were found in the $\delta_{cor.}(O_2/N_2)$ time series across 2018, so that we successfully corrected the fractionation of $O_2$ and $N_2$ both for C-130H and C-130R". Unfortunately, details of the air sampling line from the inlet to flask sampler have not been informed to researchers from Japan Ministry of Defense, which makes it difficult to add further discussion.

Line 153: A value of 1.35 for fossil fuel OR is not given in Keeling and Manning (2014) or in Keeling (1988) which is referenced therein, where is this value from?   Typically, the value used for the weighted global average for fossil fuel consumption is higher than this
Lines 170-172: We have corrected the global OR for fossil fuel combustion to 1.37 calculated based on fossil emission by category summarized in GCP (Friedlingstein et al., 2020).

Figures 4 and 5: I think the scale differences between panels (a) and (b) in each of these figures needs to be explained explicitly in the methods section when discussing NICAM-TM. I would also suggest adding to the figure caption to note that the x-axis scales differ
Lines 172-173, Figs. 4 and 5 caption: The sentence has been added to denote the scale difference between the observed and simulated data, as suggested.

Figure 5: Add reference to different altitudes in the figure caption e.g. observed in the troposphere over MNM at various altitudes
Figure 5 caption: The figure caption has been modified, as suggested.

Line 155-158: I would suggest further explaining what is meant by ignoring the dAM(APO),

particularly as this is frequently referred back to in the results/discussion, and I don't think this sentence fully explains this

Lines 176-182: The sentences have been added to explain what is meant by ignoring the $\delta^{AM}$(APO), as suggested.

Line 175: To avoid confusion I would suggest referring to "the figure" by figure number, it is not immediately clear which figure you are referring to as in the previous sentence you referred to both figures 4 and 5.

Line 214: The words "the figure" have been changed to "Fig. 5(a)", as suggested.

Figure 8: figure caption states "relative to the corresponding values at 6 km" but in text it says "relative to surface values"?

Figure 8 caption: The words "values at 6 km" in the caption have been corrected to "surface values", as suggested.

Line 235: Why are these values from figure 9(b) relative to the corresponding values at 6 km, but the values in figure 8(b) are relative to the surface? If there is no reason for this, I would suggest being consistent between the figures

Figure 9 caption: The words "corresponding values at 6 km" have been changed to "corresponding values at 1.3 km".

Figure 10: the scale size for the bottom panel (12 per meg a^-1) is smaller than that for the top and middle (-14 per meg a^-1. I would suggest changing this so they are visually comparable

Figure 10: The scale sizes have been adjusted to make them easy to compare visually.

Line 301: why has 1.37 been used as the OR here, but 1.35 above?

We have corrected the global OR for fossil fuel combustion to 1.37 throughout the paper, calculated based on fossil emission by category summarized in GCP (Friedlingstein et al., 2020).

technical corrections
Line 52: change "artificial fractionation on O2/N2" to "artificial fractionation of O2/N2"

Line 52: The words "artificial fractionation on $O_2$/$N_2$" have been changed to "artificial fractionation of $O_2$/$N_2$", as suggested.

Line 69: I don't think western should be capitalised here, should read "western North Pacific"

Line 69: The word "Western" have been changed to "western".

Line 71:"heigh-altitude" to "height-altitude", or "altitude-latitude" as you have referred to altitude throughout the text

Line 71: The typo "heigh-latitude" have been corrected to "height-latitude".

Line 126: change detail to detailed

Line 143: The word "detail" has been changed to "detailed".

Line 130 – 132: this sentence is hard to comprehend due to the number of and's, I suggest rewording
Line 132: Change have to has

Lines 147-149: The sentence has been rewritten, as suggested.

Line 262: change to "is a global average"

Line 307: The words "as a global average" have been changed to "is a global average".

Line 285: change Fig. 12 to Fig. 11

Line 331: "Fig. 12" has been changed to "Fig. 11".

Line 298: Pg C, here and elsewhere

We recognize the unit you suggested is more familiar with our research field, however, I have used this considering the Editor's comment.

Line 431 and 440 : Formatting of references is not consistent, for all other references publication tear is at the end of the reference. These two references also say "and co-authors", rather than having a full author list which should be present

Lines 487-498: The formats of the references have been corrected. Thank you for pointing that out.

Other changes

Lines 146-147: The sentence has been modified and Tohjima et al. (2005) has been added to reference since we have noticed that we used $X_{O2}$ of 0.2094 in their study to calculate the observed $\delta$(APO).

Lines 201-203, Figs. 4 and 5: The sentence has been added to note the data selection in the digital filtering technique, and the observational data deviated from the best-fitted curves more than $\pm 3\sigma$ have been excluded from Figs. 4 and 5.

Figure 8 caption: The sentence to show the method to calculate the amplitude of seasonal APO and

$CO_2$ cycles have been added.

---

## Author Response (AR2)

**Responses to the Editor**

Comments to the author:

Dear Dr Ishidoya

many thanks for your revised submission to ACPD.

I believe that you have addressed most of the reviewers' comments satisfactorily (with a few exceptions as noted below).

However, I've found an error in one of the coefficients in Eq. 10 and have a few other concerns with respect to the presentation, calculations and use of units. To give you sufficient to carry out the necessary corrections, I have indicated that major revisions are necessary even if the time required may well be shorter than the corresponding time allowance.

Best regards

Jan Kaiser

Editor ACP

Thank you very much for your significant and useful comments on the paper "Spatiotemporal variations of the $\delta(O_2/N_2)$, $CO_2$ and $\delta(APO)$ in the troposphere over the Western North Pacific" by Ishidoya et al. We have revised the manuscript, considering your comments and suggestions. Details of our revision are as follows. The line numbers denote those of the revised manuscript.

1) ACPD asks for the SI System of Units the Recommendations of the IUPAC Green Book to be followed. Amount fractions should therefore be reported using their SI unit "mol mol-1" or derivatives thereof (e.g., "μmol mol-1"). This also avoids potential confusion between "ppm" and "per meg". There are various instances in the text and the axis labels of Figs. 4 to 7 and 9 where "ppm" needs to be replaced with "μmol mol-1".

Throughout the paper: The word "ppm" has been changed to "μmol mol$^{-1}$", as suggested.

2) There are a couple of errors in equation 7:

a) The factor "x 10^6" after y(CO2) needs to be deleted.

b) The final term (2000) needs to be divided by 10^6.

c) Also, amount fractions are not reported in "per meg", so "in per meg" should be removed from l. 144.

Lines 147-148: The errors in the equation have been corrected, as suggested.

3) The dry amount fractions should be written with the species enclosed in parentheses, i.e., X(N2) and X(O2), in line with the IUPAC Green Book and the practice you have adopted for other symbols in this paper.

Throughout the paper: The dry amount fractions, including $X(O_2)$ and $X(N_2)$, have been written with the species enclosed in parentheses, as suggested.

4) There is an inconsistency between delta(APO) in Eq. 7 and Eq. 8. Looking at Eq. 1, the first term in parentheses on the right-hand side of Eq. 8 needs to be divided by y(N2)/X(N2) to match the corresponding term in Eq. 7. I acknowledge that there may be differences between measurement and modelling reference scales, but at least the underlying mathematical relationships of the quantity definitions should be the same.

5) Again, the first term on the right-hand side of Eq. 10 needs to be divided by y(N2)/X(N2). Also, the coefficient in the final term (for TB) should be alpha_B, not alpha_F. Please check whether your calculations based on this equation are correct.

Lines 165-181: The sentences have been rewritten as follows, considering your comments.

"Model-based changes in $\delta(APO)$, $CO_2$ amount fraction and $\delta(O_2/N_2)$ ($\Delta\delta(APO)$, $\Delta y(CO_2)$ and $\Delta\delta(O_2/N_2)$) were calculated using the following equations (e.g. Nevison et al., 2008; Tohjima et al., 2012) in per meg, µmol mol$^{-1}$ and per meg, respectively:

$$\Delta\delta(APO) = \left(\frac{\Delta y^{SA}(O_2)}{X(O_2)} - \frac{\Delta y^{SA}(N_2)}{X(N_2)}\right) + \left(\frac{-\alpha_F \cdot \Delta y^{FF}(CO_2) + \alpha_B \cdot \Delta y^{FF}(CO_2)}{X(O_2)}\right) + \frac{\alpha_B \cdot \Delta y^{OC}(CO_2)}{X(O_2)}, \quad (8)$$

$$\Delta y(CO_2) = \Delta y^{FF}(CO_2) + \Delta y^{OC}(CO_2) + \Delta y^{TB}(CO_2), \quad (9)$$

$$\Delta\delta(O_2/N_2) \cong \left(\frac{\Delta y^{SA}(O_2)}{X(O_2)} - \frac{\Delta y^{SA}(N_2)}{X(N_2)}\right) + \frac{-\alpha_F \cdot \Delta y^{FF}(CO_2)}{X(O_2)} + \frac{-\alpha_B \cdot \Delta y^{TB}(CO_2)}{X(O_2)}, \quad (10)$$

where $\Delta y(O_2)$, $\Delta y(N_2)$ and $\Delta y(CO_2)$ are changes in dry amount fractions of the respective gases calculated using NICAM-TM. The superscripts "SA", "FF", "OC" and "TB" denote the seasonal anomaly of the air-seas $O_2$ and $N_2$ flux, $CO_2$ flux from fossil fuel combustion, ocean and terrestrial biosphere, respectively. $X_{O2}$ and $\alpha_B$ have the same meaning as in Equation (7), while $X(N_2)$ is the dry amount fraction of $N_2$ in the atmosphere and $\alpha_F$ is the global average OR for fossil fuel combustion. In this study, we adopted $X(N_2) = 0.7808$ and $\alpha_F = 1.37$. The $\alpha_F$ was calculated from the fossil fuel and cement production emissions by fuel type for the period 2012-2019, reported by GCP (Friedlingstein et al., 2020), and the oxidative ratios for the different fuel type were taken from Keeling (1988). It should be noted that we assume initial amount fractions of $y(O_2)$ and $y(N_2)$ are equal to $X(O_2)$ and $X(N_2)$, respectively, in eqs. (8) and (10). We can rewrite equation (8) as:

$$\Delta\delta(\text{APO}) = \Delta\delta^{SA}(\text{APO}) + \Delta\delta^{FF}(\text{APO}) + \Delta\delta^{OC}(\text{APO}). \quad (11)$$

Finally, $\delta(O_2/N_2)$, $y(CO_2)$ and $\delta(\text{APO})$ obtained from NICAM-TM are reported as respective deviations of $\Delta\delta(\text{APO})$, $\Delta y(CO_2)$ and $\Delta\delta(O_2/N_2)$ from arbitrary reference points, in other words, reported on different scales from those of observations."

5) Again, the first term on the right-hand side of Eq. 10 needs to be divided by y(N2)/X(N2). Also, the coefficient in the final term (for TB) should be alpha_B, not alpha_F. Please check whether your calculations based on this equation are correct.

Eq. 10: The coefficient in the final term has been changed to $\alpha_B$. This is just a literal error, and our actual calculations of $\Delta\delta(O_2/N_2)$ are correct.

6) The "C" in "CO2" in Eqs. 8 and 10 should be written in upright font.

Eq. 8 and 10: We have revised them according to your instruction.

7) l. 22: Please split this sentence into a readable form, e.g., "annual net sea-to-air marine biological O2 flux during El Niño and net air-to-sea flux during La Niña." Parentheses should be used for clarification, not saving space (cf. Alan Robock's EOS piece, "Parentheses Are (Are Not) for References and Clarification (Saving Space)", https://eos.org/opinions/parentheses-are-are-not-for-references-and-clarification-saving-space).

Line 22: The sentence has been modified, as suggested.

8) l. 32: The equal sign "=" and the unit "mol mol-1" should be deleted. If you want to emphasise that you are referring to an amount ratio, this should be done by an appropriately name quantity, e.g., "oxidative amount ratio", but I don't think that is necessary.

Line 32: The equal sign "=" and the unit "mol mol$^{-1}$" have been deleted.

9) l. 43 and elsewhere: Height should be replaced with altitude. Altitude is the vertical distance between an object and sea level. For an aircraft, this would be 100s or 1000s of metres. In contrast, the height of an aircraft would typically be of the order of metres.

Throughout the paper: The words "height" have been changed to "altitude", as suggested.

10) l. 87: I think the position of the word "only" is not as intended. Do you mean to say, "we only use Teflon tube in the upstream ..." (i.e., not elsewhere)? Or do you mean to say that you only use it to avoid absorption? Part of the reason I am asking is because Teflon performs much worse in terms of O2 absorption and permeability than glass or steel. Also, Langenfelds (PhD Thesis, 2002) reported PTFE (Teflon) o-rings to have the worst performance for O2/N2 stability in storage experiments.

Lines 87-89: The sentence has been modified to make the meaning clearer as "It is noted that we only use Teflon tube in the upstream of the diaphragm pump to avoid absorption and/or permeation of O2 due to a pressurization of the Teflon tube".

Was January 2016 the only flight when you used Teflon as tubing material downstream of the pump? What tubing material did you use downstream of the diaphragm pump during other flights? If you don't have that information, please explain why.

Lines 91-92: The sentence "During other flights, we use a flexible tube made of stainless steel in the downstream of the diaphragm pump" has been added.

11) l. 96: Please add the specific DOI (cf. section "Data availability").

Lines 98-99: We have added the specific DOI.

12) l. 118: The unit should be written "per meg (per meg)^-1".

Lines 120 and Fig. 2: The words "per meg per meg$^{-1}$" has been changed to "per meg (per meg)$^{-1}$".

13) l. 131: Please split this sentence into a readable form.

Lines 133-135: The sentence has been modified, as suggested.

14) l. 146: Which value of X(O2) did you use for your calculations?

Lines 151: We have added the information.

15) l. 190: How do you explain the much lower variability in delta(Ar/N2) data for periods of C-130R aircraft use? It seems odd that a mere a change of aircraft type should lead to such a reduction. Were there differences in the inlet or sampling system? If you don't have that information, please explain why.

Lines 77-79, 84-87 and 202-206: Details of the air sampling line from the inlet to flask sampler have not been informed to researchers, so that it is difficult to explain the cause(s) of the much lower variability in $\delta(Ar/N_2)$ data for periods of C-130R. The visible difference is the location of the air-conditioning blowing nozzles of C-130H and C-130R. The former and the latter are attached at the roof in the cockpit and at the front of the assistant driver's seat, respectively, and air sampling tubes are inserted into the nozzles.

16) l. 199: This description is a bit too sparse to reproduce what you have done. Please include more detail on what filter parameters you used, e.g., the cut-off periods, filter orders, etc. When you write, "fundamental and first harmonics", do you mean two frequencies? Or a linear change (secular trend)

and just one frequency (presumably annual)?

Lines 209-215: The sentences have been rewritten to add the description of the digital filtering technique.

17) l. 237 & elsewhere: Please avoid abbreviations such as "w/o". Also, it would be clearer to always write the expression "without-SH-flux run" with hyphens.

Throughout the paper: The word "w/o" has been changed to "without-SH-flux run", as suggested.

18) l. 279, Figs. 8b and 9b: The text refers to 6 km, but the values in Fig. 9b are relative to 1.3 km. More importantly, the queries from referee 3 about the different reference altitudes (panels 8b and 9b) and, I'd like to add, also the reference latitudes (panels 8a and 9a) have not been addressed.

Lines 264 and 294, Figs. 8a, 8b, 9a and 9b: The text has been corrected to refer to 25.5°N and 1.3 km. The figures and related captions have been modified to address the referee's and your queries.

19) l. 280, 395, 396, 399: Is NICAMT-TM different to NICAM-TM, or is this a typo? If not a typo, please explain the differences between NICAMT and NICAM.

Throughout the paper: The typo "NICAMT-TM" has been corrected to "NICAM-TM".

20) l. 311: Why 0.84? The solubility of $O_2$ is about 0.91 times the solubility of Ar.

Line 326 and Fig. 11: The word "0.84" has been changed to "about 0.9", and we have updated Fig. 11 using the coefficient of 0.9. As you know, the coefficient changes slightly depending on sea water temperature.

21) l. 332 & 345: "yr" should be replaced with "a".

Lines 347 and 360: The word "yr" has been changed to "a".

22) Caption Figs. 4 & 5: Please include a reference describing how you obtained the "best-fit" curves and secular trends.

Caption Figs. 4 & 5: The reference "Nakazawa et al. (1997)" has been added to the captions.

Finally, the font size of the labels in Figs. 2 to 10 is too small to be easily legible. Please increase it to match at least the size of the main text (when the figure is scaled to the size intended for reproduction).

Figs. 2 to 10: The font size of the labels have been increased. If it is still small, please point them out.